

# A comprehensive insight into trajectory climatology and spatiotemporal distribution of dust aerosols in China

Lu Yang[1], Lu She[1], Yahui Che[2], Jiayu Zhang[1], Zixian Feng[1], Chen Yan[1]

[1]School of Geography and Planning, Ningxia University, Yinchuan, 750021, China

[2]School of Engineering and Built Environment, Griffith University, Brisbane, 4111, Australia

*Correspondence to*: Lu She (shelu_whu@nxu.edu.cn), Yahui Che (yahui.che@griffithuni.edu.au)

**Abstract.** Airborne dust aerosols impact negatively the climate, ecosystems, air quality, and human health. To mitigate these impacts, it is crucial to identify their three–dimensional spatiotemporal distribution, transport pathways and driving factors. In this study, the three–dimensional spatiotemporal variations and distribution of dust aerosols in China from 2007 to 2021

were first analyzed using multiple dust datasets, including Modern Era Retrospective Analysis for Research and Applications version 2 (MERRA–2) dust aerosol optical depth (DAOD) data, ultraviolet aerosol index (UVAI) data from the Ozone Monitoring Instrument (OMI), and the Vertical Feature Mask (VFM) product of Cloud–Aerosol Lidar with Orthogonal Polarization (CALIOP). Also, the transport pathways and potential source regions for dust haze in principle provincial capital cities of West and North China in spring were identified using the Hybrid Single–Particle Lagrangian Integrated

Trajectory (HYSPLIT) model and Potential Source Contribution Function (PSCF). Additionally, DAOD variations over different land cover types and the impacts of meteorological driving factors were discussed with geographic detectors. Results indicate that: (1) The multi–year average of DAOD in China from 2007 to 2021 was 0.076. A mutation in annual average DAOD occurred between 2010 and 2011, with an insignificant increasing trend between 2007–2010, a downward trend between 2011–2021 and a significant downward trend between 2014–2017. (2) The Taklamakan Desert exhibited the

highest DAOD for the entire China during spring throughout the years, with DAOD values ranging from 0.4 to 0.6 and UVAI values exceeding 2.0. The highest frequency of dust occurrence in the northwest and northern regions is at an altitude of 2–4 km in spring and summer, and of 0–2 km in autumn and winter, while it is at an altitude of 4–6 km for the Qinghai–Tibet region. (3) The dust transport routes for the provincial capital cities can be primarily divided into: western, northwestern, northern, southwestern, and local. Northwest cities are notably affected by dust from surrounding deserts. Dust

originating from the Qaidam Basin and the Hexi Corridor can be carried further downwind to inland cities, such as Xining and Lanzhou. A dust backflow was found in Beijing and Tianjin. Moreover, the discussion revealed that barren and cropland had the highest DAOD and additionally precipitation, evaporation, and soil moisture were identified as the strongest driving factors affecting dust aerosol variations. The combination effect of precipitation and temperature had the highest explanatory power, ranging from 0.72–0.84, followed by 0.75–0.81 for precipitation and U10m wind speed and 0.67–0.75 for

temperature and evaporation.



## 1 Introduction

The World Meteorological Organization (WMO) defines sand and dust storms as a meteorological phenomenon during which strong winds lift large amounts of mineral materials from bare and dry surface soils into the atmosphere (WMO, 2011). The airborne dust aerosols, generated during dust events, are one of the major components of the atmosphere and play a vital role in global climate change (Liu et al., 2022b). They can directly affect the radiation balance of the Earth by absorbing and scattering solar radiation and emitting long–wave radiation (Sokolik and Toon, 1996; Tegen, 2003; Huang et al., 2006a; Slingo et al., 2006) and in an indirect way by changing cloud properties as cloud condensation nuclei (Rosenfeld et al., 2001; Sokolik et al., 2001; Sassen, 2002; Huang et al., 2006b). Furthermore, dust particles that carry nutrients such as iron, nitrogen, and phosphorus, can be transported over long distances downwind to remote regions and oceans, affecting the biogeochemical cycle of land and oceans (Jickells et al., 2005; Kohfeld and Tegen, 2003; Richon et al., 2018). Meanwhile, strong winds during dust storms irreversibly damage ecosystems by causing loss of nutrients and water in the soil, exacerbating drought and land degradation (Field et al., 2010). Additionally, the high concentrations of dust aerosols can cause a drastic reduction in visibility, triggering severe air pollution (Tao et al., 2012) and human health issues including respiratory diseases, cardiovascular diseases, conjunctivitis, and skin irritations. (Goudie, 2014; Aghababaeian et al., 2021).

Dust storms, as a common natural disaster, frequently occur in the arid and semi–arid regions of China. This is because scarce water sources, loose surface soil, sparse vegetation and extremely fragile ecological environments in arid areas provide a rich material basis for the formation of dust (Guo et al., 2018; Liu et al., 2021a; Liu et al., 2023). The natural dust source areas in China include the Hexi Corridor in Gansu Province, Alxa League in Inner Mongolia, Taklimakan Desert and its surroundings in the south of Xinjiang Autonomous Region, the Qaidam Basin in Qinghai, the northern slope of Yinshan Mountains in Inner Mongolia and Otingdag Sandy Land and its surrounding areas, along the Great Wall in Inner Mongolia, Shanxi Province and Ningxia Hui Autonomous Region (Qiu et al., 2001; Xiao et al., 2008; Wang et al., 2001). Among them, the Taklamakan Desert and the Gobi Desert, located in Northwest China along "the Belt and Road", emit millions of tons of dust aerosols to the atmosphere over surrounding regions. These particles are usually transported eastward by the prevailing westerlies through China to North and South Korea, and Japan (Murayama et al., 2003; Natsagdorj et al., 2003), and sometimes are taken farther across the Pacific Ocean to North America (Duncan Fairlie et al., 2007; Huang et al., 2008).

Quantitative aerosol satellite remote sensing technology is expected to overcome the inherent limitations of ground–based networks, providing long–term broad coverage datasets to capture global and regional variations as well as the transport of dust aerosols (Baddock et al., 2021; Chen et al., 2023a). Remote sensing datasets can be primarily divided into three types, including the early developed the absorbing aerosol index based on ultraviolet wavelengths, aerosol optical depth (AOD) at visible wavelengths, and vertical feature measurements (VFM) using satellite based lidars. Strictly verified AOD datasets based on various satellite sensors have been widely used for regional and global dust studies. For example, Filonchyk (2018)



confirmed the predominance of dust particles during a storm, with high AOD more than 1.0, using the MODerate–resolution
Imaging Spectroradiometer (MODIS) AOD dataset. Moreover, dust AOD (DAOD), namely the AOD for dust aerosols, can
be retrieved using AOD and the simultaneous outputted parameters, such as Angstrom Exponent and sphericity. For instance,
DAOD has been successfully retrieved using the MODIS DB aerosol product (Ginoux et al., 2010, 2012; Pu and Ginoux,
2018).

The OMI Ultraviolet Aerosol Index (UVAI) has the advantage in distinguishing absorbing aerosols from non–absorbing
aerosols, even over bright sandy surfaces (Li et al., 2021). This capability has proven effective in exploring the
spatiotemporal variations of dust aerosols over dust source regions in China. Xu et al. (2015) conducted a detailed analysis
of the spatiotemporal variations in dust observed over Xinjiang and Inner Mongolia in Northern China from 2005 to 2008
based on OMI dust aerosol index. The results indicate that dust activities in Inner Mongolia is weaker than that in Xinjiang.
In both regions, spring is identified as the dustiest season, with the highest dust aerosol index values in April, and dust
activities weaken rapidly in summer and increase again in winter. Liu et al. (2022a) analyzed the spatiotemporal distribution
characteristics of UVAI in the Fenwei Plain based on the UVAI daily data retrieved by OMI from 2012 to 2020. The results
indicate that the overall trend of the annual average UVAI in the Fenwei Plain over the past 9 years showed two "peaks" in
2013 and 2018, respectively.


The Cloud–Aerosol Lidar with Orthogonal Polarization (CALIOP) product provides types and vertical structure of aerosols
on a global scale, which is crucial for understanding the long–range transport of aerosols (Liu et al., 2019). This is achieved
by leveraging the sensor's sensitivity to non–spherical dust particles and its capability to detect the vertical distribution of
aerosols (Ali, 2019). Liu et al. (2019) analyzed the spatiotemporal distribution characteristics of dust aerosols in East Asia
from 2007 to 2011, using CALIOP inversion data with a focus on dust occurrence frequency. The results indicate that a
typical "dust belt" originates from the dust source areas (Taklamakan Desert and Gobi Desert), reaches eastern China, Japan,
and South Korea, and finally deposits on the Pacific Ocean. Vertically, the frequency of dust occurrence peaks particularly at
about 2 km over the dust source regions. In a subsequent study, Liu et al. (2022b) extended the analysis to the Jianghan Plain
from 2006 to 2021 using CALIOP data. The seasonal variation patterns exhibit more frequent dust activities in spring and
winter, less in summer and autumn. Specifically, the dust occurrence frequency (under cloud–free conditions) could reach up
to 0.7 in spring and winter, while remaining below 0.4 in summer and autumn.

Benefiting from the integration of high quality satellite and ground–based datasets into the AOD assimilation system, aerosol
reanalysis products such as the Modern Era Retrospective Analysis for Research and Applications version 2 (MERRA–2)
has been increasingly used in dust research, with a high temporal resolution (Guo et al., 2019; Xu et al., 2020a; Li et al.,
2022). MERRA–2 simulates AOD, near–surface/column concentrations, and vertical/horizontal flux for dust aerosols,
making it appropriate for studying the spatiotemporal trends and circulation of dust aerosols during a single dust event



(Wang et al., 2023; Yao et al., 2020) and over years (Che et al., 2023). For example, Pang et al. (2021) used MERRA–2 reanalysis data to analyze the spatial distribution and trends of global dust aerosol optical depth (DAOD) from 2000 to 2017.

The results showed that the Taklamakan Desert in Western China exhibits a high DAOD value (>0.35) in spring and summer. Sun et al. (2019) examined the spatiotemporal variations of MERRA–2 DAOD over China from 1980 to 2017. They found that the annual average DAOD in China during this period ranged from 0.004 to 0.006, with the highest DAOD occurred in the desert regions in the northwest, followed by parts of Northern China close to deserts. Xu et al. (2020b) investigated the distribution characteristics of dust aerosols over the Tibetan Plateau and Taklamakan Desert from 1980–2017 using

MERRA–2 aerosols reanalysis and found that the duration, strength and area of the high DAOD region over the Taklamakan Desert have been enhanced over the past 38 years.

The Hybrid Single Particle Lagrangian Integrated Trajectory (HYSPLIT) model has been proven to be a powerful tool for identifying dust source regions and pathways. In practice, the implementation of HYSPLIT model is usually incorporated

with cluster analysis and Potential Source Contribution Function (PSCF) to identify the sources and transport pathways for a dust event at a specific observation site (Salmabadi et al., 2020; Aili et al., 2021). Chen et al. (2023a) conducted a 48 hour forward and backward trajectory clustering analysis on dust, polluted dust, polluted continental, and elevated smoke aerosols in 10 typical regions of China from June 2006 to December 2020 using the HYSPLIT model. The results indicate significant regional differences in the trajectories of different aerosol types. In the Beijing–Tianjin–Hebei region, dust aerosols

originated from the Mongolian Plateau, accounting for 57.88% of the total trajectories. Bao et al. (2022) simulated the airflow trajectories of dusty weather at three meteorological stations (Dalanzadgad, Erlian, and Beijing) with the HYSPLIT model from March to June 2016–2020 for 72 hours before and after 1000 m height. The results show that the dusty weather at the three meteorological stations is mainly caused by winds transporting the northwest dust to the downwind stations, accounting for about 65.5% of the total path. Yang et al. (2021) examined the potential sources of dust in 8 cities in China

from 2015 to 2020 using PSCF. In addition, a large number of studies on single dust storms or dust storms at some stations over a long timespan using HYSPLIT can be found for China (Filonchyk et al., 2019; Aili et al., 2021; Aili et al., 2023; Bao et al., 2023; Ye and Zheng, 2023).

By far, existing dust studies in China have primarily focused on long term spatiotemporal distribution of dust activities

indicated by MODIS DAOD (Han et al., 2022; Song et al., 2020), MERRA–2 (Sun et al., 2019; Pang et al., 2021), CALIPSO DAOD and VFM (Proestakis et al., 2018; Huang et al., 2015b), and OMI UVAI (Guo et al., 2017; Xu et al., 2015). These studies examine the overall levels of dust activities in China and sub–regions such as North western China and Qing–Tibet Plateau, using multiple satellite datasets. Notably, the long term transport characterises of dust based on metrological observations have been rarely included. While the widely–used HYSPLIT model has been applied in studies on

dust trajectories in or around desert regions and even in remote downwind cities in China, most of them have focused on a single or several target site or cities. This study aims to address this gap by investigating the long–term transport



characteristics of dust in principle provincial cities in China. A five–years trajectory climatology of dust trajectory from 2017–2021 is established for these selected cities using the HYSPLIT model, to deeply understand the transport patterns and potential sources of dust events in these urban areas. The study also offers the long term three–dimensional spatiotemporal

distribution of dust aerosols from 2007 to 2021 based on multi–source data, including traditional remote sensing (OMI), satellite–borne lidar (CALIPSO) and model reanalysis data (MERRA–2), providing insights into dust activities in China and the three sub–regions. Objectives include:

(1) To calculate the annual and seasonal variations and trends of DAOD in four sub-regions and the whole of China from 2017 to 2021 using MERRA-2 aerosol reanalysis;

(2) To map the seasonal distribution of dust aerosols over the same period using MERRA-2 DAOD and OMI UVAI;

(3) To identify the seasonal frequency of dust events at different altitudes over the same period using the CALOP VFM product;

(4) To build a trajectory climatology of dust aerosols for selected cities and dust source regions from 2017 to 2021 using the HYSPLIT model;

(5) To conduct a cluster analysis and a PSCF analysis on the sources of dust for the selected cities and typical dust source regions.

## 2 Study Area and Data

### 2.1 Study Area

Figure 1 shows the land cover of the study area in 2021, which is derived from MODIS Collection 6.0 land cover product

(MCD12C1) (https://ladsweb.modaps.eosdis.nasa.gov (accessed on 10 December 2022)) (Mark and Damien, 2015). Due to the environmental impacts governed by high pressure in the Pacific Ocean, Indian Ocean, and Siberia, China, located in the eastern part of the Asian continent, is one of the world's dust storm hotspots (Knippertz, 2014; Bao et al., 2023). To facilitate a comprehensive analysis, the mainland China has been divided into four sub–regions: the Northwest, Qinghai–Tibet, the North, and South (Fig. 1), according to. This classification is based on their geographic locations, land covers, and climate

conditions, aligning with the four major geographic divisions in China characterized by geographic location, natural geography, and cultural geography (Xie et al., 2004). The Northwest and Qinghai–Tibet regions are predominately covered with barren and grassland, the North region with croplands, and the South regions with forests and shrublands. Main natural dust source regions, such as the Taklamakan Desert and Gobi Desert, are primarily located in the Northwest region (Yan et al., 2002; Chen et al., 2017; Li et al., 2023). The insufficient precipitation in this region caused a reduction in vegetation

cover, subsequently contributing to enhanced dust storms in selected years (Wang et al., 2004). This intricate interplay between land cover, climate conditions, and geographical features highlight the significance of understanding regional dynamics in the context of dust storm occurrences in China.



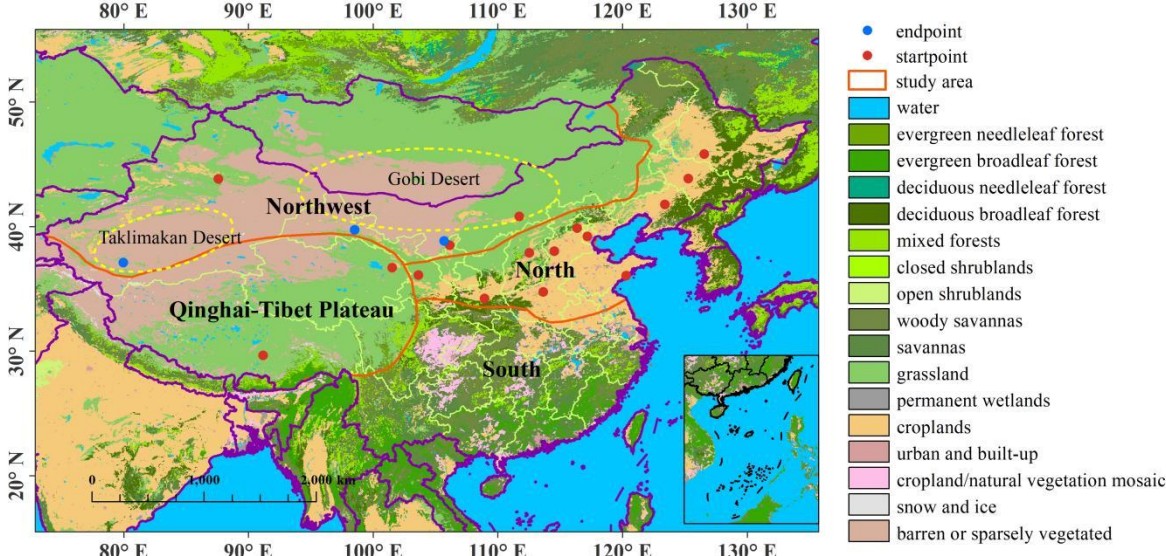

**Figure 1: Land cover map of the study area, with the distribution of PM sites (The red dots represent the endpoint of the backward trajectory, and the blue dots represent the starting point of the forward trajectory).**

## 2.2 Datasets

### 2.2.1 Satellite Data

(1) OMI OMAEROe UVAI

The UVAI dataset in the OMI/Aura Multi–wavelength Aerosol Optical Depth and Single Scattering Albedo L3 1 day Best Pixel in 0.25 degree × 0.25 degree V3 (OMAEROe) Level–3 product from 2007–2021 was selected to investigate the seasonal spatial pattern of dust aerosol in this study (Debora and Pepijin, 2012). The OMAEROe algorithm uses the full UV–to–visible spectral coverage of OMI to derive spectral aerosol extinction optical depth (Torres et al., 2007). OMI UVAI is retrieved spectrally at 342.5 and 388 nm, with a global coverage. Compared to other satellite aerosol products, OMAEROe UVAI has strengths in identifying weakly absorbing aerosols and strongly absorbing aerosols and also characterizing dust aerosols in diverse environment, including the Qinghai–Tibet plateau.

(2) CALIOP VFM product

The CALIPSO satellite, launched jointly by NASA and the National Centre for Space Studies (CNES) of the United States on 28 April 2006 (https://www-calipso.larc.nasa.gov/ (accessed on 20 December 2022)), was designed to investigate the impact of clouds and aerosols on Earth's radiation balance and climate. The main instrument on board CALIPSO is the CALIOP, a dual–wavelength (532 nm and 1064 nm) Polarized Lidar that allows observation of the vertical structure of clouds and aerosols on a global scale. The CALIOP VFM product provides the vertical distribution and classification



information of clouds and aerosols (Huang et al., 2015a). In this product, aerosols are classified into six types, including dust, polluted dust, smoke, clean continental, polluted continental, and clean marine (Omar et al., 2009). The spatial resolution of the VFM product decreases as altitude increases. Specifically, the horizontal resolution is 333 m below 8.2 km and 1 km between 8.2 and 20.2 km, while the vertical resolution is 30 m below 8.2 km and 60 m between 8.2 and 20.2 km (Winker et al., 2019; Deng et al., 2010).

The CALIOP product has strengths in determining the severity of dust storms, with a reliable capability to measure the height and composition of the dust plumes (Fissehaye, 2020; Li et al., 2023). Mielonen et al. (2009) verified CALIOP with 36 AERONET globally. The results showed that the best agreement (91%) was achieved with coarse absorbing dust aerosols. In this study, the CALIOP Level 2 VFM product from 2007 to 2021 was used to identify the frequency of dust event occurrence and the three–dimensional distribution of dust aerosols in different seasons.

(3) MODIS IGBP land cover product

This study used the MODIS International Geosphere–Biosphere Programme (IGBP) land cover dataset spanning from 2007 to 2013 in MCD12C1 (https://lpdaac.usgs.gov (accessed on 13 May 2023)). This dataset is produced annually from 2001 to the present, with a spatial resolution of 0.05o×0.05o (Friedl and Sulla–Menashe, 2022). The MODIS IGBP dataset encompasses 17 land covers, including Evergreen Needleleave Forests (ENF), Evergreen Broadleaf Forests (EBF), Deciduous Needleleave Forests (DNF), Deciduous Broadleaf Forests (DBF), Mixed Forests (MF), Closed Shrublands (CSH) Open Shrublands (OSH), Woody Savannas (WSA), Savannas (SAV), Grasslands (GRA), Permanent Wetlands (WET), Croplands (CRO), Cropland/Natural Vegetation Mosaics (CRV), Permanent Snow and Ice (PSI), Urban and Built–up Land (URB), Barren (BRN) and Water Bodies (WAT) (Friedl et al., 2002). To simply the IGBP dataset, the 17 land cover types were grouped into 9 general categories, including croplands (CRO, CRV), forests (ENF, EBF, DNF, DBF, MF), grasslands (GRA), shrublands (CSH, OSH, WSA, SAV), barren (BRN), water bodies (WAT), urban and built–up land (URB), permanent wetlands (WET), and permanent snow and ice (PSI) (Wu et al. 2023). Six main land cover types (croplands, forests, grasslands, shrublands, barren, and urban and built–up land) were selected to analyze DAOD variations over different land covers.

**2.2.2. MERRA–2 Reanalysis data**

The monthly MERRA–2 DAOD dataset from 2007 to 2021 was selected to analyze the spatiotemporal variations of DAOD. Additionally, corresponding metrological datasets including wind speed, temperature, evaporation, soil moisture, surface pressure, U10 m wind speed, and V10 m wind speed datasets for the same period in MERRA–2 meteorological reanalysis were also selected to discuss the meteorological driving factors influencing the distribution and variations of DAOD in China (https://disc.gsfc.nasa.gov/ (accessed on 1 February 2023)). The MERRA–2 meteorological reanalysis products provide a variety of meteorological parameters from 1980 to the present. It incorporates an upgraded version of the Goddard



Earth Observing System Model Version 5 (GEOS–5) and a data assimilation system (Version 5.12.4) (Molod et al., 2015), enabling it to assimilate data from the latest microwave sounders and hyperspectral infrared radiation instruments. Moreover, the MERRA–2 aerosol reanalysis includes AOD, surface/column concentration of five aerosol components: sea salt, sulfate in (SO4 and SO2), organic carbon, dust, and black carbon (Randles et al., 2017; Che et al., 2022). Notably, among these aerosol parameters, only AOD dataset is constrained by satellite and AERONET AOD data (Randles et al., 2017).


### 2.2.3. Particulate Matter (PM) Data

PM concentration data was sourced from the national real–time urban air quality release platform of the National Environmental Monitoring Station of China (http://www.cnemc.cn/ (accessed on 22 November 2022)). The hourly PM concentration data from 19 stations (Table 1) with sufficient data in provincial capital cities in the Northwest, northern, and

Qinghai Tibet region during 2017–2021 were used to identify the dust weather processes according to the Ambient Air Quality Standard (GB3095–2012) of China. The South region was excluded because of its considerable distance from natural dust sources and a prevailing long–term presence of non–dust aerosols.

A transport trajectory clustering analysis and a potential source area analysis were conducted during spring dust weather at

PM sites in 16 provincial capital cities (Urumqi, Yinchuan, Hohhot, Lhasa, Xining, Lanzhou, Xi'an, Taiyuan, Zhengzhou, Shijiazhuang, Beijing, Tianjin, Qingdao, Shenyang, Changchun, Harbin) in the northwest, Qinghai–Tibet, and northern areas. Based on previous studies (Wang et al., 2004; Wang et al., 2015; Zhao et al., 2015; Guan rt al., 2019; Bao et al., 2023), the main potential dust sources in China include the Taklamakan, western Inner Mongolia, and Hexi regions. Therefore, with the highest vulnerability to dust events, cities located in or close to these dust sources were also included in Table. 1. As a result,

three cities located in the main dust source regions, including Hotan in the Taklamakan region (Shao et al., 2011), Jiuquan in the Hexi region (Xu et al., 2020a), and Alxa league in the Badain Jilin Desert (Li et al., 2019), were selected for analyzing the dust transport pathways for dust originating from these main dust sources.

**Table 1. Geolocations of PM stations in selected provincial cities.**

| City | Longitude(°E) | Latitude(°N) | City | Longitude(°E) | Latitude(°N) |
|---|---|---|---|---|---|
| Urumqi | 87.58 | 43.83 | Yinchuan | 106.14 | 38.50 |
| Hohhot | 111.73 | 40.81 | Lhasa | 91.18 | 29.65 |
| Xining | 101.52 | 36.69 | Lanzhou | 103.63 | 36.10 |
| Xi'an | 108.94 | 34.23 | Taiyuan | 112.52 | 37.89 |
| Zhengzhou | 113.64 | 34.75 | Shijiazhuang | 114.53 | 38.02 |
| Beijing | 116.37 | 39.87 | Tianjin | 117.18 | 39.21 |
| Qingdao | 120.30 | 36.21 | Shenyang | 123.41 | 41.77 |





| Changchun | 125.28 | 40.81 | Harbin | 126.56 | 45.82 |
| Hotan | 79.95 | 37.12 | Alxa League | 105.70 | 38.84 |
| Jiuquan | 98.50 | 39.73 | | | |

**3 Methods**

**3.1 Mann–Kendall (MK) mutation test**

The Mann–Kendall (MK) mutation test method is a widely used analysis technique for detecting mutation in time series data, particularly in the hydrological and atmospheric fields (Guan et al., 2021; Duan et al., 2022). This approach does not require data linearity or a normal frequency distribution, becoming one of the time series analysis techniques that WMO

recommends (Amini, 2020; Baghbanan et al., 2020; Libiseller and Grimvall, 2002). For a time series with n samples, the rank sequence $S_k$ is defined as the cumulative sum of $r_i$ (Eq. 1).

$$S_k = \sum_{i=1}^{k} r_i, (k = 2, 3, ..., n), \; r_i = \begin{cases} 1, x_i > x_j \\ 0, x_i \leq x_j \end{cases} (j = 1, 2, ..., i) \tag{1}$$

where $r_i$ is a flag that represents the relationship of $x_i$ and $x_j$ (the value immediately preceding $x_i$ in the time series). Assuming that the time series ($x_i$, i = 1, 2, ..., n) is randomly independent, its order statistics ($UF_k$) can be calculated by the following equation:

$$UF_k = \frac{S_k - E(S_k)}{\sqrt{Var(S_k)}} \; (k = 1, 2, ..., n) \tag{2}$$

where $UF_1 = 0$; $E(S_k)$ and $Var(S_k)$ are the mean and variance of the cumulative number $S_k$, respectively.

$$E(S_k) = \frac{k(k-1)}{4}, Var(S_k) = \frac{k(k-1)(2k+5)}{72} \tag{3}$$

Time series x was set in the reverse order as $x_n, x_{n-1}, ..., x_1$. Above steps were repeated while enabling $UB_k = -UF_k$, (k = n, n − 1, ..., 1) and $UB_1 = 0$.

In this study, the significance level α = 0.05, the critical value $\mu_\alpha = \pm 1.96$. Subsequently, the $UF_k$ and $UB_k$ curves, and

±1.96 straight lines are plotted on the same graph. The interpretation of the results is based on the behaviour of these curves. If the value of $UF_k$ is greater than 0, the sequence is an upward trend in the time series, otherwise it indicates a downward trend. A significant upward or downward trend is denoted by the two sequence curves exceeding the critical line. If the $UF_k$ and $UB_k$ curves intersect and the intersection point is between the critical lines, the intersection point signifies the beginning of the mutation.



## 3.2 Identification of dust weather

In order to effectively identify the dust weather processes, the start and end times of dust events were determined according to the "Supplementary Provisions for Air Quality Evaluation in Cities Affected by Dust Weather Processes" (Wang et al., 2022; Yang et al., 2021). The criteria for the start time of a dust event are: PM10 greater than 150 $\mu g \cdot m^{-3}$ and one of the following conditions are met:

(1) Hourly PM10 concentration is greater than or equal to twice the average PM10 concentration in the preceding six hours;

(2) The ratio of hourly PM2.5 concentration to PM10 is less than or equal to 50% of the average ratio of the preceding six hours.

The criteria for the end time of the dust event are:

(1) Hourly PM10 concentration first drops to a relative deviation of less than or equal to 10% from the average PM10 concentration in the six hours before dust weather.

## 3.3 HYSPLIT model

The trajectory computational model HYSPLIT (http://ready.arl.noaa.gov/HYSPLIT.php), developed by the National Oceanic and Atmospheric Administration (NOAA), was used to analyze the transport of dust in China during the spring of 2017–2021. The model has been widely used in research on the transport and diffusion of atmospheric pollutants (Baker, 2010; Ge et al., 2017; Wang et al., 2014). In this study, HYSPLIT model was used to generate air mass backward and forward trajectories, tracing the dust transport at a height of 500 m above the ground during dust events. The meteorology input for the HYSPLIT model was the reanalysis data from the National Centers for Environmental Prediction (NECP) of the United States with a horizontal resolution of $1.0° \times 1.0°$. For the backward trajectory analysis, the 48–hour backward trajectories ending at Urumqi, Yinchuan, Hohhot, Lhasa, Xining, Lanzhou, Xi'an, Taiyuan, Zhengzhou, Shijiazhuang, Beijing, Tianjin, Qingdao, Shenyang, Changchun, Harbin (red dots in Fig. 2) were calculated for determining the dust sources for the capital cities of the provinces in Northwest, northern and Qinghai Tibet regions. For the forward trajectory analysis, the 72–hour forward trajectories starting at Hotan, Alxa league, Jiuquan (blue dots in Fig. 2) were calculated for tracking the dust transport in three typical dust source regions, namely, the Taklamakan Desert, Badain Jilin Desert, and Hexi regions, respectively.

## 3.4 Potential source contribution function (PSCF)

The PSCF method has been widely used to evaluate the transport pathways of pollutants and the contribution of potential source areas (Cheng et al, 2017; Meng et al., 2020; Zhao et al., 2023). PSCF is a conditional probability function that uses trajectories to calculate and determine the spatial distribution of potential source areas. In detail, $PSCF_{ij}$ is defined as the ratio





of the number of contaminated trajectory endpoints ($m_{ij}$) for a specific grid (ij) to the number of trajectory endpoints ($n_{ij}$) of all trajectories passing through this grid (ij):

$$PSCF_{ij} = \frac{m_{ij}}{n_{ij}} \tag{4}$$

When the airflow retention time within the grid is short (with a small $n_{ij}$ value), PSCF will contain a large uncertainty. In order to reduce this uncertainty, a weight function $W_{ij}$ is introduced to weight it, defined as:

$$W_{ij} = \begin{cases} 1.00, \, n_{ij} > 80 \\ 0.70, \, 20 < n_{ij} \le 80 \\ 0.42, \, 10 < n_{ij} \le 20 \\ 0.05, \, n_{ij} \le 10 \end{cases} \tag{5}$$

The product of the weigh factor $W_{ij}$ and the PSCF value can be expressed as WPSCF, which suggests the probability of

occurrence of the weighted pollution trajectories.

$$WPSCF = W_{ij} \times PSCF_{ij} \tag{6}$$

### 3.5 Optimal parameters–based geographical detector (OPGD) model

The geographic detector model was first developed by Wang and Xu (2017), aiming to identify the spatial heterogeneity and reveal the driving forces behind. The key aspect of using geographic detectors is to determine the optimal scale for spatial hierarchical heterogeneity through spatial data discretization. However, a traditional geographic detector needs to set

parameters manually when discretizing the driving factors, which inevitably results in strong subjectivity in the results (Song et al., 2020). To address this issue, the Optima parameters–based geographical detector (OPGD) model was used to reveal the driving factors of DAOD spatial distribution heterogeneity and improve the accuracy of spatial data analysis (Song et al., 2020). Meteorological factors such as precipitation, wind speed, temperature, evaporation, soil moisture, ground pressure, U10 m wind speed, and V10 m wind speed were selected to analyze their impact on DAOD through factor detection and

interaction detection.

 (1) Factor detection

Factor detection was used to detect the effect of each factor on the spatial heterogeneity of the dependent variable Y (DAOD) with the following equation:

$$q = 1 - \frac{\sum_{h=1}^{L} N_h \sigma_h^2}{N\sigma^2} \tag{7}$$

where q is the explanatory power of a factor on the spatial heterogeneity of DAOD, normally taking the value of [0,1]. A larger q indicates stronger explanatory power of the factor on the spatial heterogeneity of DAOD. L represents the total number of layers of influencing factors. $N_h$ and N represent the number of units in layer h and the entire region, respectively, $\sigma_h^2$ and $\sigma^2$ are the variance of the Y values for layer h and the entire region, respectively.



(2) Interaction detection

Interaction detection was used to quantify the combined effect of different factors on the explanatory power of the analysis variables, including nonlinear–weaken, uni–weaken, bi–enhance, independent, and nonlinear–enhance. The interaction between two driving factors were measured by the power of determinant (q) value and the interactive q value ($q(x1 \cap x2)$).

## 4 Results

### 4.1 Temporal variations in dust aerosols

Figure 2 shows the annual average MERRA–2 DAOD for the entire Mainland China and four sub–regions (left) and for the entire China (middle) and the MK mutation test for MERRA–2 DAOD in the entire Mainland China (right) over the period 2007–2021. The multi–year DAOD mean in China during 2007–2021 was 0.076, showing an overall slight downward trend with a slope of –0.0002 (Fig. 2b). Notably, in nine years China experienced annual average DAOD values exceeding the multi–year mean for the periods of 2007–2011 and 2018–2021. The annual average DAOD peaked in 2009 and 2018, exceeding the multi–year averages by 7.1% and 9.2%, respectively. The MK Mutation test for the annual average DAOD values in China from 2007–2021 (Fig. 2c) shows that both UB and KB curves for DAOD and their intersection fall within the confidence level of 0.05. The mutation is identified between 2010 and 2011, as indicated by the intersection of the curves. This is also reflected by an insignificant increasing trend during 2007–2010, a downward trend during 2011–2021 and a significant downward trend during 2014–2017.

The four regions of China also show significant differences in DAOD (Fig. 2a). The DAOD value in the northwest region shows a fluctuating downward trend from 2007 to 2016, reaching its lowest value in 2016 followed by a fluctuating upward trend, and continues to remain peak values of around 0.18. Similar trends are observed in other regions, with peak and low years corresponding to those of the Northwest region. The lows for the North and Qinghai–Tibet Plateau are of 0.048 and 0.040 in 2016, respectively. Notably, the North region experienced DAOD peaks in 2009, while the Qinghai–Tibet and South regions experienced their peaks in DAOD in 2008.



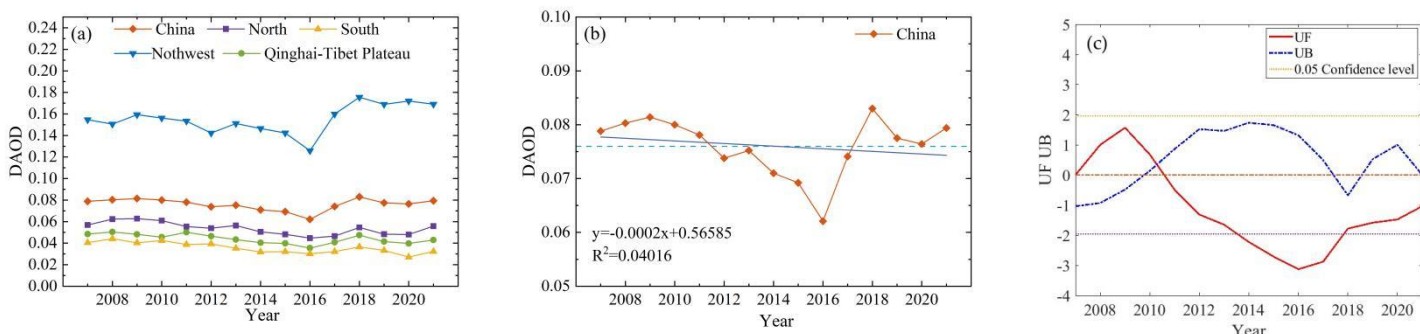

**Figure 2: Annual average MERRA–2 DAOD in the entire Mainland China and four sub–regions (a), for the entire China with more details (b) and MK mutation test for MERRA–2 DAOD in the entire Mainland China over the period 2007–2021.**

Figure 3 shows the inter–annual variation of the seasonal DAOD in China and the four sub–regions over 2007–2021.

Seasonal means were obtained as averages of monthly aggregates for spring (Mar–May), summer (Jun–Aug), autumn (Sep–Nov) and winter (Dec–Feb). Overall, Spring is identified the dustiest season in China as well as four regions, with higher DAOD than other seasons. The Northwest region is identified as the most susceptible region to dust aerosols, experiencing constantly high DAOD through Spring to Autumn since 2012 (Fig. 3a). Notably, differences in DAOD among these three seasons have been gradually decreased over the study period, suggesting that the severity of dust storms in the Northwest

region of China has been increased over the past 15 years. This increased severity in dust aerosols in the Northwest region also influenced the entire China, as showed by similar DAOD variation with the Northwest region in four seasons (Fig. 3a). In Qinghai–Tibet, DAOD was significantly lower than in the Northwest (Fig. 3c), while spring and summer are similarly identified the dustiest seasons. In northern and southern China, high DAOD only occurred in Spring (Fig. 3d and 3e). Different from other regions, DAOD exhibits a significant decreasing trend in these two regions in the dusty season over the

past 15 years.

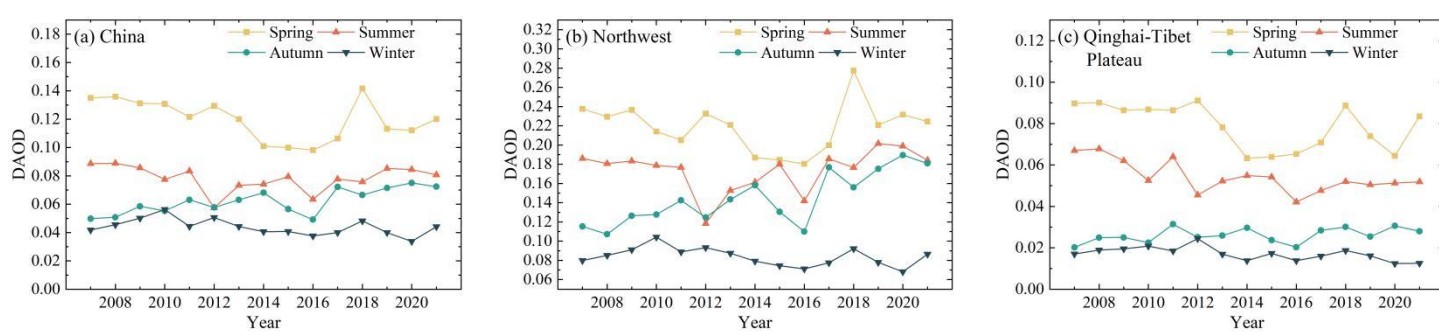



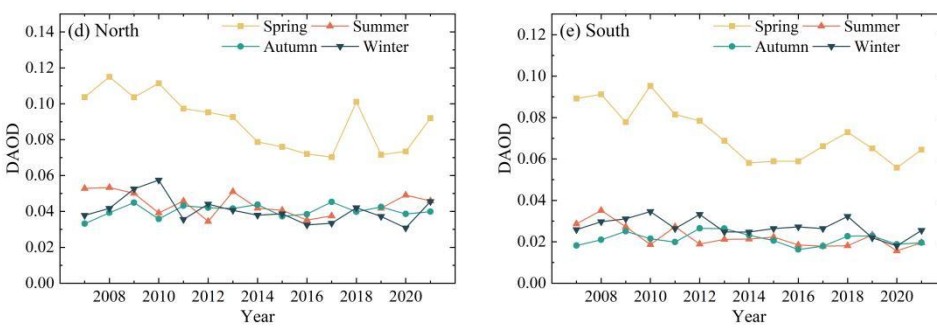

**Figure 3: Seasonal variation in MERRA–2 DAOD over China and four regions over the period 2007–2021.**

The seasonal average DAOD for four sub–regions and China over the 15 years are given in Table 2. In all four sub–regions, the seasonal DAOD values follow a consistent pattern: Northwest > North > Qinghai–Tibetan > South. Spring DAOD was significantly higher than the other seasons, with the multi–year seasonal DAOD of 0.219, 0.090, 0.079, and 0.072 in the
Northwest, North, Qinghai–Tibetan, and South, respectively. As the main natural dust source region, the Northwest region shows much higher DAOD levels than other regions, even that in winter is at the same level of severity for other regions in dustiest spring. The Qinghai–Tibetan shows a feature of two dusty seasons in a year: the dustiest spring with a DAOD of 0.079 and the second dustiest summer with a DAOD of 0.054. The one–dusty season regions, the South and North, exhibit relatively low DAOD (around 0.021 to 0.044) in other seasons.

**Table 2. Seasonal average of DAOD in various regions over the years.**

|  | China | Northwest | Qinghai–Tibetan | North | South |
|---|---|---|---|---|---|
| Spring | 0.120 | 0.219 | 0.079 | 0.090 | 0.072 |
| Summer | 0.078 | 0.174 | 0.054 | 0.044 | 0.022 |
| Autumn | 0.062 | 0.144 | 0.026 | 0.040 | 0.021 |
| Winter | 0.044 | 0.084 | 0.017 | 0.041 | 0.027 |
| mean | 0.076 | 0.155 | 0.044 | 0.054 | 0.036 |

**4.2 Spatial variation of dust aerosols**

Figure 4 shows the spatial variation of annual and seasonal variations in MERRA–2 DAOD (Fig. 4a, 4c, 4e, 4g, and 4i) and OMI UVAI (Fig. 4b, 4d, 4f, 4h, and 4j) in China over the period of 2007–2021. Regions with large natural dust sources are
typically with high DAOD. The Taklamakan Desert in Xinjiang is the most important contributor to the highest DAOD in Northwestern China, which ranges from 0.4–0.6 in spring. Meanwhile, the DAOD value in western Inner Mongolia and regions along the Hexi Corridor were also relatively high, at around 0.3. High DAOD values were observed over Qaidam





sandy land in Qinghai–Tibet in spring, constituting a major contribution to the high DAOD over Qinghai–Tibet region. In southern and northern China, DAOD is relatively small on average, with relatively high DAOD occurring mainly in spring

close to the Tibetan Plateau. The spatial distribution of UVAI is similar to that of DAOD, confirming the accuracy of MERRA–2 DAOD in indicating the spatial distribution of dust aerosols.





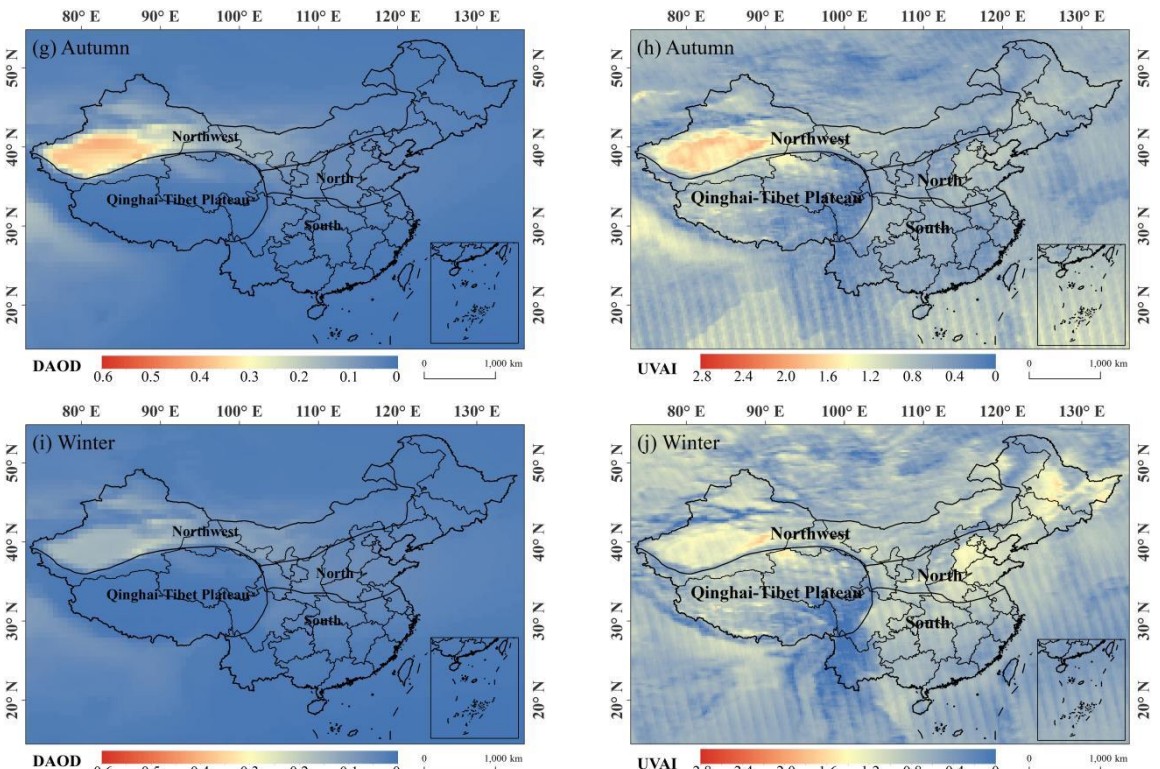

**Figure 4: Spatial variations of annual and seasonal from MERRA–2 DAOD and OMI UVAI during the period 2007–2021.**

Figure 5 shows the seasonal frequency of dust aerosols using CALIOP VFM products from 2007 to 2021. The frequency of dust occurrence exhibits distinct seasonal variations. Overall, the frequency of dust decreases sequentially from spring to

winter. The height of dust uplift is the highest in the spring, leading to transport over the longest distance downstream and thus covering the widest area. The dust frequency in spring and summer shows high similarities: it increases with altitudes, reaches the peak at altitudes of 2–4 km, and decreases at altitudes over 4 km in the entire China. In autumn and winter, the dust frequency peaks at altitudes of 0–2 km and even reaches higher than 0.6 in autumn.

From the perspective of the four sub–regions, the frequency of dust in the northwest region is higher than the other three regions at altitudes of 0–4 km. The Taklamakan Desert and its surrounding areas have the highest dust frequency (with a spring dust frequency greater than 0.4). This high dust frequency is related to unique topography in Xinjiang region. The Taklamakan Desert is surrounded by the Kunlun Mountains in the south and the Tian Shan Mountains in the north, which is conducive to the long–term floating and maintenance of dust aerosols in the atmosphere (Tsunematsu et al., 2005). Dust

events also occur frequently in the Hexi Corridor and the Gobi Desert (with spring dust frequency ranging from 0.2 to 0.4). Notably, no dust is found from 0–4 km in each season in the Qinghai Tibet mainly because the average elevation region is



about 4 km (Spicer et al., 2003). Thus, most of dust events in the Qinghai Tibet region were observed at altitudes of 4–6 km, with a value of around 0.2 and less at an altitude of 6–8 km in spring.

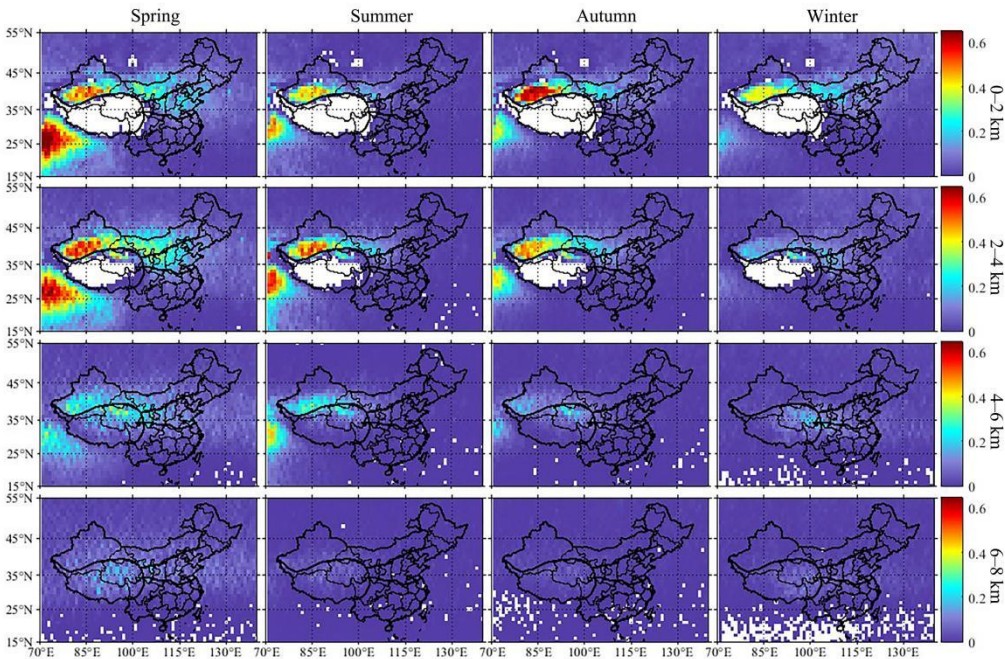

**Figure 5: Seasonal average dust frequency distribution in China from CALIPSO data during the period 2007–2021.**

**4.3 Transport pathways and potential sources of dust aerosols**

**4.3.1 HYSPLIT clustering analysis**

Based on the GDAS meteorological data provided by NCEP, the 48 h backward trajectories for selected cities were calculated using HYSPLIT, as well as a cluster analysis was performed using the outputted backward trajectories (Fig. 6). The results show that Urumqi has two air mass trajectories originating from the desert of Kazakhstan: one follows a westward path and the other follows a northwest path. The highest proportion of air mass trajectories for Urumqi originates from the Gurbantunggut Desert, which is the largest contributor to dust events in Urumqi due to the short distance (Fig. 6a). Yinchuan has three north-westward air mass trajectories and additionally two northward path remotely from Mongolia and shortly from the Mongolia. 31.81% of the air mass trajectories originate from the Ulan Buh Desert in Inner Mongolia, which is close to Yinchuan (Fig. 6b). Hohhot is mainly influenced by the air masses passing through Mongolia, as well as those from the Mu Us Desert with a short distance (Fig. 6c).

The Qinghai Tibet Plateau has a dry climate with an elevation over 4 km (Feng et al., 2020), which is conducive to the transport of dust. All three main air mass trajectories for Lhasa originate from the overseas, and 61.33% originating from the south of Lhasa (Fig. 6d). A long trajectory of the air mass transported from the Southern Xinjiang Basin to Xining indicates



the dust aerosols in Xining could be sourced from remote desert areas thousands of kilometers away. The trajectory of air masses from the Qaidam Basin accounts for 25.90%. The shortest trajectory of the easterly path indicates that the air mass moves slowly or stays in place from the east under the influence of calm or mild easterly winds (Fig. 6e) (Tan et al., 2022). Meanwhile, the terrain of the Huangshui River Basin is not conducive to dust transport and diffusion, with a small pressure
difference and a stable atmosphere (Tan et al., 2022).

Due to its large population, the North region is predominated by croplands and dust is usually transported from remote desert regions in the Northwest. The trajectory of air mass with the highest proportion in Lanzhou originates from the Tengger Desert and passes through the Hexi Corridor, which has been one of the most frequent areas of sandstorms in China due to
the exposed surface conditions and frequent strong winds (Zhao et al., 2015). This is because Lanzhou is located in the downwind region of these two regions (Fig. 6f). 38.35% of the air mass trajectory in Xi'an originates from the border between Xinjiang and Mongolia, passing through the Alxa Plateau, Ningxia, and Gansu. Another trajectory originates from a southward direction close to Xi'an (Fig. 6g). The main trajectory of the air mass in Taiyuan is the northwest path, with 87.95% of the air mass trajectory flowing from Mongolia through the Mu Us sandy land (Fig. 6h). Zhengzhou (Fig. 6i),
Shijiazhuang (Fig. 6j), Beijing (Fig. 6k), and Tianjin (Fig. 6l) share the high similarities in source areas for the air mass trajectories, which originate from remote regions in Mongolia and Russia and also in their near dry lands in China. In addition to the influence of upstream transportation, dust backflow is also common as the air mass trajectory moves from south to north. The sources for air mass trajectories in Qingdao (Fig. 6m), Shenyang (Fig. 6n), Changchun (Fig. 6o), and Harbin (Fig. 6p) are all clustered into three regions, including two in Russia and one in the North China Plain.

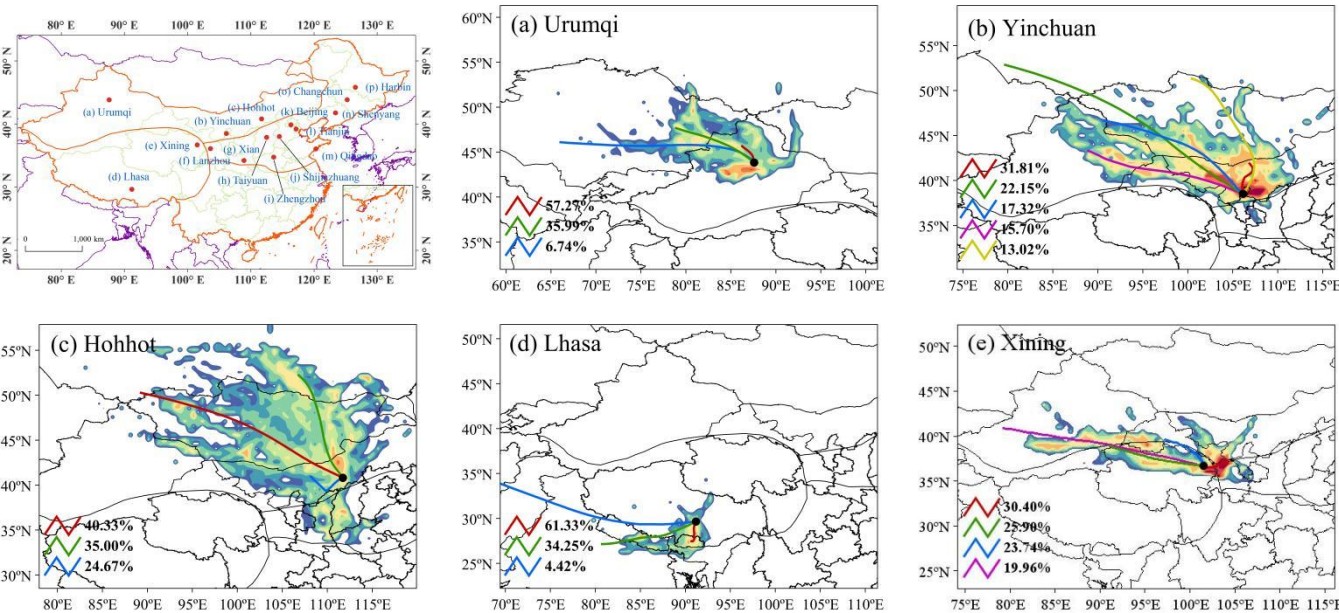





**Figure 6: Cluster analysis and WPSCF value distribution of cities in the northwest, Qinghai–Tibet, and northern regions during spring dust weather from 2017 to 2021.**





### 4.3.2 PSCF Analysis

Colors in Fig. 6 correspond to the WPSCF values, indicating potential dust sources in various cities. The higher the WPSCF value, the greater the probability that the area is a potential dust source. A value between 0.4 and 0.6 is defined as "relatively high", between 0.6 and 0.8 as "high", and 0.8 or above as "extremely high".

Regarding the provincial capital cities in the Northwest, the most potential dust source tends to be the nearest large natural deserts. From the spatial distribution of WPSCF values, the most potential source area of dust in Urumqi is the Taklamakan Desert. High WPSCF values for Yinchuan City are mainly concentrated in the Alxa Plateau, Mu Us Sandy Land, and Hunshandak Sandy Land in western Inner Mongolia and the extremely high WPSCF is located in the Kubuqi Desert. The potential source areas are widely distributed with no obvious high WPSCF values. Notably, most of them are located in desert regions.

The provincial capital cities in the Qing–Tibet Plateau shows similarities in potential dust sources. The extremely high WPSCF value areas for Xining are located in the nearest Tengger Desert to Xining, and the high WPSCF values are found to the south of Lhasa. This indicates the potential dust sources for these two cities are the nearest deserts or dry lands.

Provincial capital cities in the North are situated away from large natural deserts, while most of them are located in the downwind areas of northwestern deserts in China or Mongolian/Russian deserts. The potential dust source for Lanzhou seems located around the border between Gansu, Inner Mongolia, and Ningxia, with relatively high WPSCF values. As Xi'an is located in the downwind area of the Hexi Corridor, the WPSCF values along the Hexi Corridor is relatively high. High WPSCF value areas for Taiyuan are distributed in Hebei, Inner Mongolia, and Shaanxi. The potential sources for Zhengzhou and Shijiazhuang are located in the western and northeastern of Inner Mongolia. Beijing shares the same potential dust sources with Tianjin, with high WPSCF values in the neighboring Hebei as well as Inner Mongolia. High WPSCF values in Qingdao, Shenyang, Changchun, and Harbin are mainly distributed around the trajectory of air masses, suggesting that the dust is primarily from remote dust source regions.

### 4.3.3 Analysis of transportation paths in typical dust source areas

The forward air mass trajectory shows that 52.81% of dust originating from Hotan was only transported a short distance and deposited in Xinjiang region. 21.79% of the air mass trajectories suggest that dust aerosols from the Taklamakan Desert can reach and accumulate in the northern slope of the Qinghai Tibet Plateau. Westward trajectories also indicate dust can be transported to the west with a short distance. Notably, the Hotan dust can be transported to the Hexi Corridor and further to Liaoning and North Korea via Northern China. Most of the air mass trajectories originating from Alxa League, which is close to the end of the Hexi Corridor, show that the majority of dust from Alxa League dust is transported to the end of the





Hexi Corridor. 42.18% can reach western Shandong via Ningxia, Shanxi, and Henan. The transportation trajectory for dust from Jiuquan is similar to that from Alxa League, with the highest proportion of dust transported to the end of the local Hexi Corridor. The dust from Jiuquan passes through Ningxia and can finally reach Henan. Additionally, 13.63% of dust can be
carried over long–distance from Inner Mongolia through Hebei, Liaoning, and Jilin to the west coast of the Pacific Ocean.

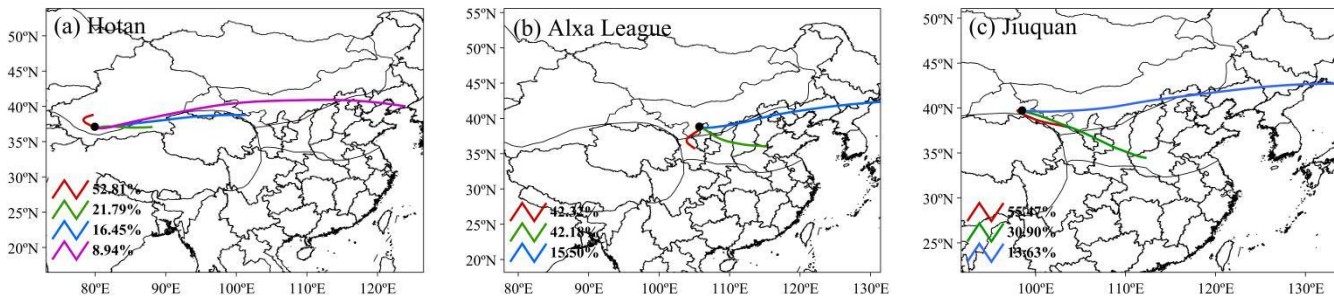

**Figure 7: Cluster analysis of cities during spring dust weather from 2017 to 2021.**

## 5 Discussion

The spatiotemporal distribution of dust aerosols mapped in this study aligns with previous studies. Existing studies consistently suggest that China exhibits high levels of DAOD over dust source and nearby areas, such as the Taklamakan Desert, western Inner Mongolia, along the Hexi Corridor, and the Qaidam Basin (Pang et al., 2021; Xu et al., 2020b). For
example, over the Taklamakan Desert, which is identified as the Chinese largest natural dust source region (Qiu et al., 2001), the highest DAOD was constantly observed throughout the entire year. Regarding DAOD over the entire China, a decreasing trend was observed, consistent with the analysis of DAOD in East Asia conducted by Wang et al. (2024). However, differences are found in terms of DAOD peaks in vertical direction. The frequency of dust occurrence was identified to reach its peak depending on the season. Specifically, it peaked at 2–4 km in summer and 0–2 km in autumn and winter. This
differs from Liu et al. (2019) who found that the peak frequency of dust occurrence occurred about 2 km above the dust source area.

The trajectory climatology of dust aerosols constructed in this study has significantly complemented the existing studies in this field, which can be classified into two categories: for a) large natural deserts and b) for populated downwind cities.
Firstly, Yu et al. (2019) built a trajectory climatology for the Taklamakan Desert (2001–2011) and the Gobi Desert (2001–2003) using MISR data and the HYSPLIT model. Unlike their study that focuses on deposition points of each dust plumes originating from two selected deserts during the periods, this study identified primary dust sources for principle provincial cities. Similar studies of trajectory climatology as Yu et al. (2019) can be found for other natural deserts, such as the Tarim Basin (Gao and Washington, 2009) and Inner Mongolia (Tan et al., 2012). Furthermore, this study has also added existing



body of knowledge on trajectory climatology of dust for a downwind station in China, such as Xi'an (Yang et al., 2021),
       Beijing (Zhu et al., 2011), and Wuhan (Liu et al., 2022).

       In complementation to DAOD analysis conducted four sub–regions, this study further discussed the variations in annual
       mean DAOD over six main land covers (croplands, forests, grasslands, shrublands, barren, and urban and built–up land)
from 2007 to 2021 (Fig. 8). The DAOD values on forests and shrublands are the lowest, with multi–year average DAOD
       values of 0.035 and 0.033, respectively. In contrast, the DAOD value on Barren is the highest, with a multi–year average of
       0.148, followed by croplands (0.061). In the western part of the northwest region and the northern part of the Qinghai Tibet
       region, the primary landcover is barren lands, which are characterized by drought, lack of water, and low vegetation
       coverage, combined with factors such as overgrazing and unreasonable reclamation, resulting in frequent spring dust storms
(Wang et al., 2021a; Wang et al., 2021b). In the northern region is croplands, with flat terrain and no natural barriers to
       prevent wind and sand, water on the surface of the land evaporates rapidly, causing the soil to become dry and prone to dust
       generation (Liu et al., 2020). Overall, the annual average DAOD values for different land covers follow a descending order:
       barren > croplands > grasslands > urban and built–up land > forests > shrublands. Notably, the mean DAOD for the six land
       covers all decreased to their lowest in 2016, which is consistent with the annual change trend of DAOD in Fig. 2b. During
the study period, the DAOD values on barren, croplands, and grasslands showed an upward trend, while the DAOD values
       on the other three land covers showed a downward trend. These temporal variations highlight the dynamic relationship
       between land cover types and changes in DAOD.

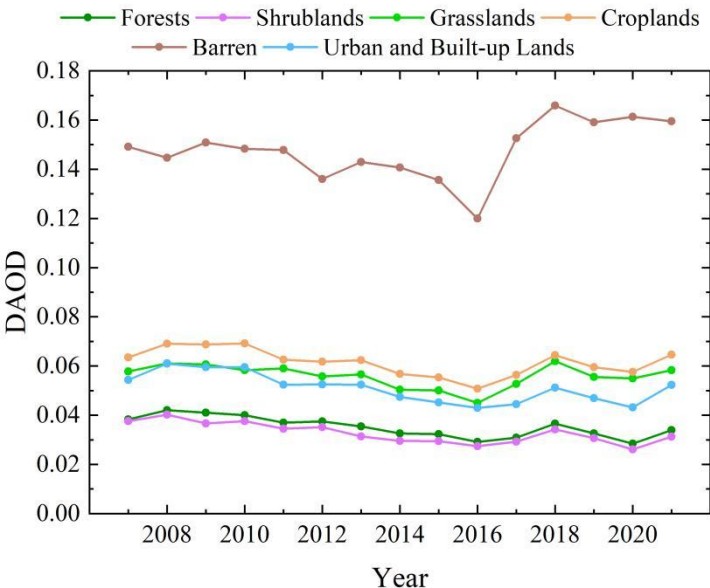

**Figure 8: Annual average DAOD values on eight major land use types from 2007–2017.**



The factor detector revealed the degree of influence of meteorological driving factors on DAOD over China. The effect of each driving factor on DAOD was extracted by calculating the q–value for each factor (Fig. 9). All driving factors exhibited statistically significant effects on DAOD, passing the 1% significance test. The order of significance of each variable affecting DAOD based on the q–value is as follows: evaporation (x4) > soil moisture (x5) > precipitation (x1), with an explanatory power of more than 40%. It should be noted that temperature (x3) has a strong explanatory power on the

changes in DAOD, while wind speed (x2), surface pressure (x6), U10m wind speed (x7), and V10m wind speed (x8) have less explanatory power in the context of DAOD variations.

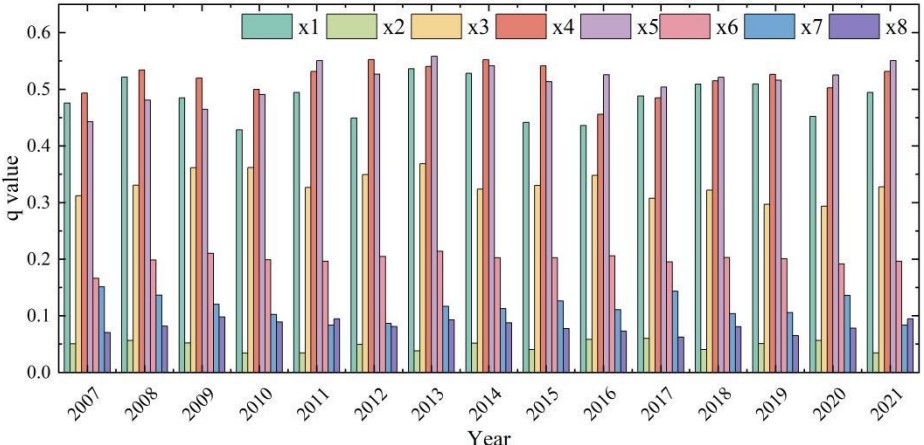

**Figure 9: Contribution of single variables to changes in DAOD studied by the factor detector.**

The interaction between various driving factors manifests as bi–enhance or nonlinear–enhance. The explanatory power (q

value) of any two driving factors on DAOD is stronger than that of a single factor. Precipitation (x1)∩temperature (x3) = (0.72–0.84) has the highest explanatory power, the second highest for precipitation (x1)∩(U10m wind speed) x7 = (0.75–0.81), and the third is temperature (x3)∩evaporation (x4) = (0.67–0.75). In addition, the interaction results of precipitation (x1)∩wind speed (x2), precipitation (x1)∩soil humidity (x5), precipitation (x1)∩surface pressure (x6), wind speed (x2)∩evaporation (x4), temperature (x3)∩soil humidity (x5), evaporation (x4)∩soil humidity (x5), evaporation (x4)∩surface

pressure (x6), evaporation (x4)∩U10m wind speed (x7), and soil humidity (x5)∩U10m wind speed (x7) were all greater than 0.6. It indicates that DAOD variations are a result of multiple factors.





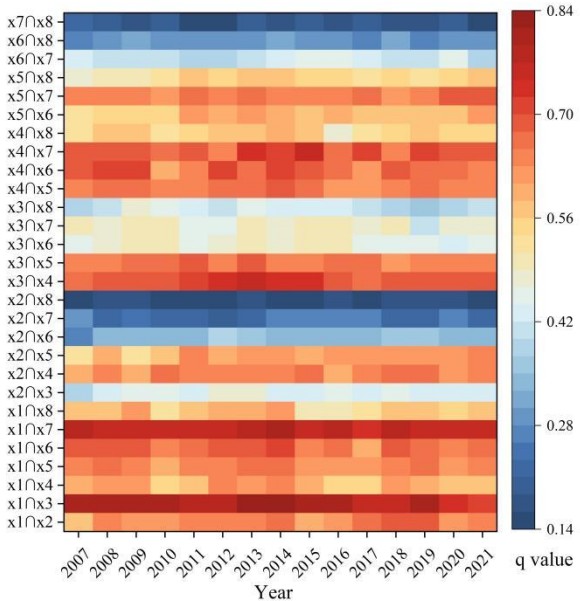

**Figure 10: Interaction of various factors on DAOD.**

## 6 Conclusions

In contrast to previous studies that exclude long–term large coverage dust transportation, this study built a five–year trajectory climatology for principle provincial cities in China for providing insights into long–term dust transportation and the corresponding potential dust sources in China. Spatiotemporal variations of dust aerosols in China using aerosol reanalysis and remote sensing products from multiple sources was also included to enhance the understanding of regional dynamics of dust aerosols and additionally establish connections with previous studies on spatiotemporal distribution of dust

aerosols in China. To achieve this, MEAAR–2, OMI, and CALIPSO aerosol products were selected to explore the spatiotemporal variation characteristics of dust aerosols in four sub–regions of China from 2007 to 2021 at annual and seasonal scales. Simultaneously, the five–year trajectory climatology was built using the HYSPLIT model for selected cities. To further examine the sources of dust, cluster analysis and PSCF analysis were conducted on the sources of dust for these cities. Additionally, transportation trajectories for typical dust source areas were also investigated. The following research

conclusions were obtained.

(1) Regrading temporal variations, the levels of dust activities showed an overall downward trend in China from 2007 to 2021, with a multi–year average DAOD of 0.076. A mutation occurred between 2010 and 2011. The seasonal and annual variation trends in dust activities in the entire China show strong consistency with the Northwest region. Notably, significant

differences exist among the four sub–regions.



(2) In terms of spatial variations, the Taklamakan Desert and Hexi Corridor serves as the primary contributor to high levels of dust aerosols in northwestern China. Spring is identified as the dustiest season while dust events may last throughout the year. The peak frequency of dust occurrence in spring and summer is observed at an altitude of 2–4 km. In contrast, during autumn and winter, the frequency of dust occurrence is observed the highest at an altitude of 0–2 km. Due to the high elevation over 4 km, the frequency of dust occurrence in the Qinghai Tibet region peaks at an altitude of 4–6 km.

(3) Affected by diverse geographical conditions, the sources and transportation paths vary across different cities. Primary dust sources for cities located near or in downwind areas are identified as natural deserts in northwestern China. The dust transport routes for cities can be mainly divided into western, northwestern, northern, southwestern, and local types, depending on the relative location of the city to natural desert. For cities near or located in deserts, dust transport paths are often pointed at the closest deserts. For downwind cities, the trajectories primarily originate from remote regions in northwestern China, Mongolia and Russia.

(4) The highest severity of dust aerosols is observed on croplands and barren regions. Precipitation, evaporation, and soil moisture are identified as the strongest driving factors for layered heterogeneity affecting dust aerosol changes, with explanatory power above 0.4. The combined impact of any two variables proves to be more influential on dust aerosols than the individual effect of a single variable. As a result, the coupling of precipitation and temperature significantly influence the occurrence of dust events.

**Data availability.** CALIOP data available at the website of Atmospheric Science Data Center, NASA Langley Research Center (https://www-calipso.larc.nasa.gov/). MODIS IGBP land cover data are available at the NASA Earth Observing System Data and Information System (https://lpdaac.usgs.gov). OMI OMAEROe UVAI and MERRA–2 Reanalysis data are available at the NASA Goddard Earth Sciences Data and Information Services Center (https://disc.gsfc.nasa.gov/). Particulate Matter (PM) data are available at the National Environmental Monitoring Station of China (http://www.cnemc.cn/).

**Author contribution.** Conceptualization, LS and LY; Methodology, LY, LS, and JZ; Software, LY, ZF, and CY; Validation, LY and LS; Formal analysis, LY, LS, and YC; Writing–original draft preparation, LY and LS; Writing–review and editing, LY, YC, and LS; Funding acquisition, LS; All authors have read and agreed to the published version of the manuscript.

**Competing interests.** The contact author has declared that none of the authors has any competing interests.

**Financial support.** This research was funded by the Science and Technology Department of Ningxia (grant no. 2023AAC05022 and 2023BEG02043) and the National Natural Science Foundation of China (grant no. 42301415).



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
