# Peer review of "A comprehensive insight into trajectory climatology and spatiotemporal distribution of dust aerosols in China"

_EGUsphere, 2024_

## Author Comment (AC1)

**Response to Reviewer 1 Comments**

Dear reviewer:

Thanks very much for taking your time to review this manuscript. We really appreciate all your valuable comments and suggestions! We have carefully considered and addressed each comment in blue with the original comments in black. Changes are highlighted in yellow in the revised manuscript.

General comments:

This study uses multiple data sets of satellite retrievals to show the spatiotemporal distribution and variation of dust aerosols over China. The transport pathways and potential source regions for dust aerosols are identified using back trajectory (HYSPLIT) model and Potential Source Contribution Function (PSCF). However, there are quite a few problems with this paper. Generally, the paper could hardly be "comprehensive" and lacks of new "insights" into this problem. Additionally, there are quite a lot of errors in sentence expressions and typos which makes the paper not easy to follow. The whole paper seems to be rough and without polishing. The authors should have been more serious in preparing their paper for submission to renowned journals like ACP. The paper definitely needs major revisions to be further considered for publication.

Thank you very much for constructive comments and suggestions, and we believe the revised manuscript has been improved significantly in terms of language, structure coherence, and the abstract and method clarity.

We changed our title to avoid overstatement,

FROM

*"A comprehensive insight into trajectory climatology and spatiotemporal distribution of dust aerosols in China"*

TO

*"Insights on the trajectory climatology and spatiotemporal distribution of dust aerosols in China from multi-source data".*

We changed through the paper to avoid the words such as "comprehensive", "deeply", "strongly" and "sufficient" as we realized these are not quite scientific terms.

We also changed through the paper to improve the reliability and fix grammar issues. We have revised our abstract to below:

*"Dust aerosols are a significant atmospheric component with profound impacts on ecosystems, air quality, and human health. As one of the major sources of dust in East Asia, China, particularly its "Three-North" region, often faces the brunt of dust events. Thus, analyzing the dust's spatiotemporal dynamics, sources, and transportation paths in this region is critical for understanding regional and global changes in dust. In this study, we analyzed the four-dimensional (latitude, longitude, altitude, and time) spatiotemporal distribution of dust aerosols in China from 2007 to 2021 using multiple datasets from Modern Era Retrospective Analysis for Research and Applications version 2 (MERRA-2), Ozone Monitoring Instrument (OMI), and Cloud-Aerosol Lidar with Orthogonal Polarization (CALIOP). We further utilized the Hybrid Single-Particle Lagrangian Integrated Trajectory model and Potential Source Contribution Function to map the seasonal dust transport pathways and identify potential dust sources in the provincial capitals within the "Three-North" regions. The results show that (1) The average Dust Aerosol Optical Depth (DAOD) across China from 2007 to 2021 is 0.075, with the Taklamakan Desert consistently recording the highest DAOD values of 0.4 to 0.6 during spring. There is a notable shift in the annual DAOD between 2010 and 2011, preceded by a minor upward trend from 2007 to 2010, and followed by a downward trend from 2011 to 2021. (2) Different regions and seasons exhibit varying DAOD trends despite the overall downward trend in China. Regions further from the dust sources (e.g., North and Northeast) show a general decline in DAOD, especially during spring, indicating reduced long-range dust transport, whereas the Northwest shows a slight upward trend, notably*

*increasing in autumn. (3) Dust is lifted to higher altitudes in spring and summer, resulting in a broader transport range. In autumn, despite high dust frequency in the Northwest, it remains concentrated at lower altitudes, thus affecting less regions. (4) The sources and transportation paths of dust vary across different cities. Cities in the Northwest China are mainly affected by dust from nearby deserts, while cities in North China are influenced by the western and central parts of Gobi Desert, and the north-eastern cities mainly impacted by the eastern Gobi Desert and the Horqin Sandy Land. The findings of this study will enhance the assessment for the Three-North Shelterbelt Forest Program by using low-cost satellite and air quality data."*

We have revised the final paragraph of introduction to better introduce the motivation and objectives of the study. We changed it to below:

*"Existing dust studies in China have primarily focused on spatiotemporal distribution of dust activities using one or several of the following data sources: MODIS DAOD (Song et al., 2020; Han et al., 2022), MERRA-2 AOD (Sun et al., 2019; Pang et al., 2021), CALIPSO DAOD and Vertical Feature Mask (VFM) (Huang et al., 2015b; Proestakis et al., 2018), and OMI UVAI (Xu et al., 2015; Guo et al., 2017). Moreover, studies focusing on dust trajectories using the HYSPLIT model often selected only one or several sites or cities in China. The objective of this study is to use multiple source data to provide insights on dust sources and transport across China with a focus on the "Three-North" Region, i.e., a sandstorm belt from Heilongjiang in the east to Xinjiang in the west with eight major deserts, four major sandy areas, and vast Gobi deserts. This study first offers a long term four-dimensional spatiotemporal distribution of dust aerosols from 2007 to 2021 based on multi-source data, including traditional remote sensing (OMI), satellite-borne lidar (CALIPSO) and model reanalysis data (MERRA-2), providing insights into dust activities in China and its six sub-regions. This is followed by a seven-year (2015–2021) seasonal dust trajectory climatology established for the capital cities in the "Three-North" region using the HYSPLIT model, to understand the transport patterns and potential sources of dust*

*events in these areas. Since almost all provincial capital cities have populations exceeding 2 million, the trajectory meteorology findings would significantly contribute to improving the effectiveness of Chinese desertification combat and environmental protection projects, especially the Three-North Shelterbelt Forest Program project, benefiting the population through the improved air quality and reduced economic losses."* (Lines 124-139, pages 4-5)

We aligned the subtitles of the methods and results so that they correspond well. We have made substantial revisions to the delineation and analysis of the subregions. We now reorganized the study area into six sub-regions: Northwest China (NWC), North China (NC), Northeast China (NEC), Southwest China (SWC), Central South China (CSC), and East China (EC), as depicted in Fig. R1 (Fig.1 in revised version). The boundaries of the sub-regions correspond to the provincial boundaries. Consequently, we have made revisions to Section 2.1, which introduces the study area, as well as the analytical sections 4. These revisions ensure that the presentation and discussion are fully congruent with the newly defined sub-regional framework, providing a more coherent and accurate geographical context for our research.

[Figure]

Fig. R1: The study area (a) and the main deserts in the study area (colored in b), with the distribution of Ambient air quality monitoring stations (dots in a)

In Section 4.3, we have enriched the discussion with an in-depth examination of potential dust transportation pathways within the source regions. This enhancement broadens the scope of our trajectory analysis, extending it from the previous focus on spring seasons over five years to include data across different seasons from the full range of years, 2015 to 2021. The reason we did not analyze earlier data is that the earliest available data from the Ambient air quality monitoring stations starts from 2015. Additionally, our analysis has been intensified, with a particular emphasis on the dust sources and their trajectories in the provincial capitals of the "Three-North" region. Moreover, we have conducted a detailed analysis of the dust sources and trajectories in capital cities, considering the variations across different seasons.

Furthermore, we have revised the discussion and conclusion sections for greater clarity.

Specific comments:

1. Introduction: Line 128: "Notably, the long term transport characterises of dust based on metrological observations have been rarely included." It's too arbitrary to say this.

Response 1:

This sentence has been replaced with

*"The objective of this study is to use multiple source data to provide insights on dust sources and transport across China with a focus on the "Three-North" Region, i.e., a sandstorm belt from Heilongjiang in the east to Xinjiang in the west with eight major deserts, four major sandy areas, and vast Gobi deserts."* (Lines 128-131, page 5)

2. Line 131: "This study aims to address this gap by …" – Please be specific about what "gap" do you mean?

Response 2:

We have added more text about the benefit and significance of this study and stressed the gap in existing studies.

*"The objective of this study is to use multiple source data to provide insights on dust sources and transport across China with a focus on the "Three-North" Region, i.e., a sandstorm belt from Heilongjiang in the east to Xinjiang in the west with eight major deserts, four major sandy areas, and vast Gobi deserts. This study first offers a long term four-dimensional spatiotemporal distribution of dust aerosols from 2007 to 2021 based on multi-source data, including traditional remote sensing (OMI), satellite-borne lidar (CALIPSO) and model reanalysis data (MERRA-2), providing insights into dust activities in China and its six sub-regions. This is followed by a seven-year (2015–2021) seasonal dust trajectory climatology established for the capital cities in the "Three-North" region using the HYSPLIT model, to understand the transport patterns and potential sources of dust events in these areas. Since almost all provincial capital cities have populations exceeding 2 million, the trajectory meteorology findings would significantly contribute to improving the effectiveness of Chinese desertification combat and environmental protection projects, especially the Three-North Shelterbelt Forest Program project, benefiting the population through the improved air quality and reduced economic losses."* (Lines 128-139, page 5)

3. Line 132: A "five–years trajectory climatology of dust trajectory" seems too short to be called "long-term". Additionally, please explain why two time ranges of "2017-2021" and "2007-2021" are used instead of a uniform range (i.e., 2007-2021)?

Response 3:

Thank you very much for pointing out this issue.

We agree that "five years" is difficult to categorize as "long term". We have carefully revised our manuscript to avoid using this term.

In addition, we have expanded the original five-year period (2017-2021) to a longer seven-year period from 2015-2021 for trajectory analysis. The reason why we did not extend the trajectory analysis period further is that the earliest data available from the Ambient air quality monitoring stations starts from 2015.

4. Line 150-152: It is arguable to say "Due to the environmental impacts governed by high pressure in the Pacific Ocean, Indian Ocean, and Siberia, China, located in the eastern part of the Asian continent, is one of the world's dust storm hotspots". Additionally, this sentence is not complete!

Response 4:

This sentence has been replaced with

*"China, located in the eastern part of the Asian continent, is one of the world's dust storm hotspots (Bao et al., 2023). Based on the administrative division of China, we divide the research area into six sub-regions: Northwest China (NWC), North China (NC), Northeast China (NEC), Southwest China (SWC), Central South China (CSC), and East China (EC) as shown in Figure 1."* (Lines 142-145, page 5)

5. Line 154-156: "This classification is based on their geographic locations, land covers, and climate conditions, aligning with the four major geographic divisions in China characterized by geographic location, natural geography, and cultural geography" – Redundant but not convincing. A lot of factors are mentioned, but which one is decisive? Additionally, how do the authors "pick" out the data within the regions since the boundaries are curves?

Response 5:

In the revised manuscript, we have reclassified six sub-regions based on China's administrative divisions: Northwest China (NWC), North China (NC), Northeast China (NEC), Southwest China (SWC), Central South China (CSC), and East China (EC). The findings for each sub-region would directly provide references for desertification combat projects, such as the Three-North Shelterbelt Forest Program, and most of such projects are designed and implemented based on this classification.

We rewrote this paragraph as:

*"China, located in the eastern part of the Asian continent, is one of the world's dust storm hotspots (Bao et al., 2023). Based on the administrative division of China, we divide the research area into six sub-regions: Northwest China (NWC), North China (NC), Northeast China (NEC), Southwest China (SWC), Central South China (CSC),*

*and East China (EC) as shown in Figure 1. Figure 1 also shows the provinces in Northern China, their capital cities and other selected cities, the location of 16 Ambient air quality monitoring stations used for urban dust trajectories (red dots represent the endpoint of the backward trajectory, and the blue dots represent the starting point of the forward trajectory). The NEC and NC are predominately covered with barren and the main natural dust source regions, such as the Taklamakan Desert and Gobi Desert, are primarily located in these two regions (Yan et al., 2002; Chen et al., 2017; Li et al., 2023a). The insufficient precipitation in this region contributes to enhanced dust storms in selected years (Wang et al., 2004)."* (Lines 142-150, page 5)

1. Line 165: Figure 1 is not properly described in the main text. What are the "PM sites"?

Response 1:

We have revised the description for Figure 1 and used the "Ambient air quality monitoring stations" to replace the "PM sites". The PM concentration data used in our study are measured by Ambient air quality monitoring stations, which are managed and operated by the China National Environmental Monitoring Centre as a national-level environmental monitoring network.

We have updated the description for Figure 1 as follows:

*"Figure 1 also shows the provinces in Northern China, the provincial capitals and other selected cities, the location of 16 Ambient air quality monitoring stations used for urban dust trajectories (red dots represent the endpoint of the backward trajectory, and the blue dots represent the starting point of the forward trajectory)."* (Lines 145-148, page 5)

2. Line 190-191: "Mielonen et al. (2009) verified CALIOP with 36 AERONET globally" – please be specific about what variables are validated ?

Response 2:

We have checked the literature carefully and explained which variables were validated. *"Mielonen et al. (2009) verified the aerosol subtypes of CALIOP measurements using daily aerosol types derived from AERONET level 2.0 inversion data. The CALIOP classification identifies "dust" as coarse absorbing aerosols, "polluted dust" as mixed absorbing aerosols, "biomass burning" as fine absorbing aerosols, "marine" as coarse non-absorbing aerosols, and "clean and polluted continental" as fine non-absorbing aerosols, based on AERONET data. The results showed the highest agreement (91%) with coarse absorbing dust aerosols."* (Lines 192-197, pages 7-8)

3. Line 206-208: Why the "six" land types are chosen instead of others?

Response 3:

This revised manuscript focuses more on dust trajectories and sources regardless of land cover. As a result, we have removed the analysis of DAOD over the six land covers and thus the associated introduction texts.

4. Line 224: How do you define "sufficient data"? Please be specific!

Response 4:

We selected one Ambient air quality monitoring stations in each city, to ensure the completeness of the trajectory analysis. The selected site must provide the most complete observations during the period from 2015 to 2021. To make this clearer, this sentence has been revised:

*"These 16 stations provide the most PM observation data among all stations in their respective cities."* (Lines 213-214, page 8)

5. Line 226-227: the "south region" defined in this study still contain a large portion of areas which could be influenced by dust. Maybe too arbitrary here.

Response 5:

Yes, we agree that the South region must be affected by dust storms. However, in this study, we try to focus on trajectory analysis of dust for cities in the "Three-North" region, which contains large areas of deserts and heavily influence densely populated

cities. We also added a statement about why the "Three-North" region was selected in the Introduction:

*"The objective of this study is to use multiple source data to provide insights on dust sources and transport across China with a focus on the "Three-North" Region, i.e., a sandstorm belt from Heilongjiang in the east to Xinjiang in the west with eight major deserts, four major sandy areas, and vast Gobi deserts."* (Lines 128-131, page 5)

*"This is followed by a seven-year (2015–2021) seasonal dust trajectory climatology established for the capital cities in the "Three-North" region using the HYSPLIT model, to understand the transport patterns and potential sources of dust events in these areas. Since almost all provincial capital cities have populations exceeding 2 million, the trajectory meteorology findings would significantly contribute to improving the effectiveness of Chinese desertification combat and environmental protection projects, especially the Three-North Shelterbelt Forest Program project, benefiting the population through the improved air quality and reduced economic losses."* (Line 134-139, page 5)

6. Line 233-234: "Therefore"? How do you "select" the cities to be included?

Response 6:

Considering the high population density and the abundance of Ambient air quality monitoring stations in provincial capitals, we have chosen to focus our analysis on the provincial capital cities in the "Three-North" region. For each city, we selected the Ambient air quality monitoring stations with the most observations to perform trajectory and potential source area analysis. As for the forward trajectories, we selected 3 cities that are close to the dust source areas and with the highest frequency of dust. We also provided an explanation in section 2.2.3., see below:

*"Among these, 13 stations (red dots in Fig. 2a) serve as endpoints for the backward trajectory analysis and are situated in the provincial capitals of Northwest China (NWC), North China (NC), and Northeast China (NEC), collectively referred to as the "Three-North" region of China—a region particularly impacted by dust storms (Guan*

*et al., 2019; Bao et al., 2023). The cities in question are Urumqi, Xining, Lanzhou, Yinchuan, Xi'an, Hohhot, Taiyuan, Shijiazhuang, Beijing, Tianjin, Shenyang, Changchun, and Harbin. The remaining 3 stations (blue dots in Fig. 2a), which act as starting points for the forward trajectory analysis, are located in three representative dust source areas. Specifically, they are Hotan near the Taklamakan Desert, Jiuquan located in the Hexi Corridor, and Alxa League within the Badain Jilin Desert. Historical dust records show that Hotan in the Taklamakan region (Shao et al., 2011) and Jiuquan in the Hexi region are the cities with the highest frequency of dust occurrence (Xu et al., 2020a)."* (Lines 215-223, page 9)

7. Line 252: "Above steps were repeated while"? which steps?

Response 7:

This sentence has been replaced with

*"Similarly, the reverse order sequence $UB_k$ can be calculated by setting time series $x$ in the reverse order as $x_n, x_{n-1}, ..., x_1$ and repeating the calculation steps for $UF_k$."*
(Lines 251-252, page 10)

8. Line 254: how do you define and set the "the critical value"?

Response 8:

In the MK mutation test, the critical value $\mu_\alpha$ is used to determine whether the statistic $UB_k$ or $UF_k$ is significant, thereby determining whether there is a significant trend in the time series. This critical value $\mu_\alpha$ is defined based on significance level $\alpha$ and normal distribution.

When $\alpha = 0.05$, the critical value is usually taken as $\pm 1.96$, because in the standard normal distribution, approximately 95% of the data falls within $\pm 1.96$ standard deviations from the mean.

9. Line 262-263: Please explain why this is the best choice?

Response 9:

We used PM10 measurements to identify the dust weather, and the start and end times of dust events were determined according to the Supplementary Provisions for Air Quality Evaluation in Cities Affected by Dust Weather Processes" (Wang et al., 2022; Yang et al., 2021). This method has been widely used in dust event identification studies (Yu et al. 2024).

In comparison with normal multispectral satellite observations, ground-based PM observations can provide more accurate Particle concentration measurements and meteorological variables. When compared to the CALIPSO observation, which can provide information on aerosol types, PM data have a higher frequency of observation, which makes it less likely to miss dust events, and it can be utilized as a fixed starting or endpoint for conducting trajectory analysis.

Additionally, we have added a discussion of dust indemnification method in section 5:

*"The PM-based identification method used in this study has advantages in determining dust events lasting over 6 hours compared to other methods. Firstly, the traditional visibility-based dust identification method is very likely to be influenced by other types of aerosols. Automated visibility sensors can measure visibilities continuously and existing studies have demonstrated that dust events identified based on visibility less 5 km correspond well to the recorded dust data (O'Loingsigh, et al., 2017). However, this visibility method tended to omit dust events with visibility greater 5 km and misclassified haze as dust. For megacities such as Beijing, the most influential factor affecting the weather is haze with a high concentration of anthropogenic aerosols (Zhang et al., 2015). Secondly, high-quality satellite dust products are popular for determining dust deposition location over a large area, whereas those satellite products are often with a low observation frequency and limited by cloud cover and satellite observation width limitations. Consequently, dust events lasting less than one day are very likely to be missed. However, this study did not consider dust events lasting less than 6 hours to avoid the influence of traffic and construction activities (Yu et al., 2024). This duration criterion is expected to be reduced in future studies to include as many dust events as possible."* (Lines 582-593, page 29)

10. Line 270: only (1) end time criteria?

Response 10:

Yes, we followed the criteria given by "Supplementary Provisions for Air Quality Evaluation in Cities Affected by Dust Weather Processes" (Wang et al., 2022; Yang et al., 2021). According to this method, there are two criteria for the start time and only one criterion for the end time.

11. Line 312: what are the "number of layers of influencing factors " and "the number of units in layer h"?

12. Line 315-318: difficult to understand here!

Response 11-12:

The section about factors discussing the spatial heterogeneity of DAOD has been removed in the revised manuscript to focus more on trajectory analysis on dust.

13. *Line 338: Figure 2: part of the figure caption is missing. BTW, the organization of (a), (b), (c) is strange.*

Response 13:

Thank you for this suggestion on figure 2. We have modified the title for figure 2 and rearranged it to correspond with the text content.

"

[Figure]

(a)  (b)  (c)

Fig. R2: Annual average MERRA-2 DAOD for the six sub-regions (a), and for the entire Mainland China (b), and MK mutation test for DAOD in the entire Mainland China (c) over the period 2007–2021."

14. *Line 343-345: the relation between DAOD and dust severity is not convincing.*

Response 14:

Thank you for the reminder, we have removed the unconfirmed conclusions. we modified that sentence,

FROM

*"Notably, differences in DAOD among these three seasons have been gradually decreased over the study period, suggesting that the severity of dust storms in the Northwest region of China has been increased over the past 15 years."*

TO

*"Notably, the differences in DAOD among these three seasons gradually decrease over the study period, especially in the autumn, when there is a significant increasing trend in DAOD over the past 15 years."* (Lines 318-319, page 12)

15. Line 370-371: The validation of the MERRA-2 data should be done more rigorously. More quantitative analysis should be given instead of just the figures. Previous studies frequently point out the underestimation of MERRA-2 AOD compared with other observations, which can be also seen here.

Response 15:

Indeed, rigorous validation of MERRA-2 AOD is extremely important to support the analysis and findings of our study. However, numerous validation studies have already been conducted and documented. Therefore, instead of repeating the validation work, we have cited these existing studies to bolster the credibility and robustness of our research. This approach allows us to build on the established validation efforts and focus our resources on new analyses and interpretations. This is also because validation of MERRA-2 data is not trivial. We have added the texts on MERRA-2 DAOD as below:

*"While direct validation on MERRA-2 DAOD is scarce, evidence of its accuracy can be inferred from the validation of two parameters determining DAOD: MERRA-2 AOD and the ratio of dust emission to total emissions. Numerous validation studies on MERRA-2 AOD have been conducted in China with AERONET data as a reference (Sun et al., 2019b; Zhang et al., 2020). For example, Wang et al. (2024) validated the accuracy of the MERRA-2 DAOD at 550nm using four AERONET stations (AOE_Baotou, Beijing, Dalanzadgad, and QOMS_CAS) for the years 2000–2022. The $R^2$ for the four selected stations ranged from 0.62 to 0.81, with the RMSE ranging from 0.013 to 0.07. However, high consistency with AERONET data from these studies cannot fully validate the accuracy of MERRA-2 AOD because AERONET data has been assimilated into the MERRA-2 AOD product (Randles et al., 2017). Independent validation studies are conducted with non-AERONET photometer AOD, such as SONET (Ou et al., 2022), SIAVNET (Shi et al., 2023), CARSNET (Che et al., 2019), SKYNET (Sun et al., 2018), as well as with satellite (Ali et al., 2022) and GCM AOD (Liu et al., 2021a). These studies consistently show that MERRA-2 AOD agree well with the reference ground based AOD data. Specifically, MERRA-2 AOD correlates CARSNET AOD with an R of 0.67 and 0.70 over NWC and NC, respectively, which are slightly lower than those with AERONET (0.85 and 0.80, respectively) (Che et al., 2019). Additionally, the ratio of dust to total particles has been validated with Lidar (Gkikas et al., 2021) and AERONET products (Che et al., 2022). The results show that MERRA-2 agrees well with the reference data."* (Lines 163-177, pages 6-7)

16. Line 425: Figure 6: please use some other colors to plot the trajectories so that they can be distinguished from the shadings.

Response 16:

Thank you for this suggestion. In the revised manuscript, we have redrawn the figures and all figures are now in the same format and style. For example, the following figure shows the cluster analysis and WPSCF value distribution for Urumchi in spring (Fig. 7a in the revised version).

[Figure]

Fig. R3: HYSPLIT backward trajectory analysis and WPSCF value distribution of cities in the NWC region for the dust events from 2015 to 2021 for the four seasons.

17. Line 516: The so-called "factor detectors" does not seem to give much information about the changes in DAOD and why the factors change year by year?

Response 17:

We have removed the discussion in this section and focused on analyzing and discussing the spatial changes, transportation trajectories, and sources of dust.

18. This paper needs a thorough proofread. There are lots of typos and errors to be corrected.

Response 18:

Thank you for your comprehensive review and valuable suggestions on our paper. We take your comments regarding the typing errors very seriously and acknowledge this oversight during the paper preparation process.

We have thoroughly proofread and revised the manuscript to avoid typos and errors.

We sincerely appreciate your comments and firmly believe that through your suggestions and our efforts, this paper has become much more complete and accurate. Some examples of changes please refer to our response to your first comment.

Minor problems:

1. Line 7: It seems arbitrary to say the dust aerosol impact is "negative".

Response 1:

This sentence has been replaced

FROM

*"Airborne dust aerosols impact negatively the climate, ecosystems, air quality, and human health."*

TO

*"Dust aerosols are a significant atmospheric component with profound impacts on ecosystems, air quality, and human health. As one of the major sources of dust in East Asia, China, particularly its "Three-North" region, often faces the brunt of dust events."* (Line 7-9, page 1)

2. Line 8: Shouldn't that be 4-dimensional instead of 3?

Response 2:

Thanks for this enlightening suggestion. Indeed, it is "4-dimensional". we modified that sentence,

FROM

*"To mitigate these impacts, it is crucial to identify their three–dimensional spatiotemporal distribution, transport pathways and driving factors."*

TO

*"Thus, analyzing the dust's spatiotemporal dynamics, sources, and transportation paths in this region is critical for understanding regional and global changes in dust. In this study, we analyzed the four-dimensional (latitude, longitude, altitude, and time) spatiotemporal distribution of dust aerosols in China from 2007 to 2021 using multiple datasets from Modern Era Retrospective Analysis for Research and Applications version 2 (MERRA-2), Ozone Monitoring Instrument (OMI), and Cloud-Aerosol Lidar with Orthogonal Polarization (CALIOP)."* (Lines 9-13, page 1)

3. Line 9: "variations and distribution" – distribution and variation.

Response 3:

We agree with the reviewers' suggestions and will incorporate the recommended changes into the manuscript. This sentence has been replaced,

FROM

*"In this study, the three–dimensional spatiotemporal variations and distribution of dust aerosols in China from 2007 to 2021 were first analyzed using multiple dust datasets, including Modern Era Retrospective Analysis for Research and Applications version 2 (MERRA–2) dust aerosol optical depth (DAOD) data, ultraviolet aerosol index (UVAI) data from the Ozone Monitoring Instrument (OMI), and the Vertical Feature Mask (VFM) product of Cloud–Aerosol Lidar with Orthogonal Polarization (CALIOP)."*

TO

*"In this study, we analyzed the four-dimensional (latitude, longitude, altitude, and time) spatiotemporal distribution of dust aerosols in China from 2007 to 2021 using multiple datasets from Modern Era Retrospective Analysis for Research and Applications version 2 (MERRA-2), Ozone Monitoring Instrument (OMI), and Cloud-Aerosol Lidar with Orthogonal Polarization (CALIOP)."* (Lines 10-13, page 1)

4. Line 29-30: As "explanatory power" is not a common sense for ordinary people, listing the ranges is not useful.

Response 4:

This revised manuscript focuses more on dust trajectories and sources, so we have removed this sentence from the abstract.

5. Line 48: "a rich material basis"?

Response 5:

This sentence has been replaced

FROM

*"This is because scarce water sources, loose surface soil, sparse vegetation and extremely fragile ecological environments in arid areas provide a rich material basis for the formation of dust (Guo et al., 2018; Liu et al., 2021a; Liu et al., 2023)."*

TO

*"The scarcity of water, loose soil texture, sparse vegetation and the fragile ecology of these regions provide conducive conditions for dust storms (Guo et al., 2018; Liu et al., 2021a; Liu et al., 2023)."* (Lines 46-47, page 2)

6. Line 60: "Remote sensing datasets can be primarily divided into three types" – it may not be appropriate to divide in this way.

7. Line 61: "the early developed the absorbing aerosol index based on ultraviolet wavelengths"?

Response 6-7:

We have removed these inappropriate expressions. This paragraph has been replaced

FROM

*"Quantitative aerosol satellite remote sensing technology is expected to overcome the inherent limitations of ground-based networks, providing long-term broad coverage datasets to capture global and regional variations as well as the transport of 60 dust aerosols (Baddock et al., 2021; Chen et al., 2023a). Remote sensing datasets can be primarily divided into three types, including the early developed the absorbing aerosol index based on ultraviolet wavelengths, aerosol optical depth (AOD) at visible wavelengths, and vertical feature measurements (VFM) using satellite based lidars. Strictly verified AOD datasets based on various satellite sensors have been widely used for regional and global dust studies. For example, Filonchyk (2018) confirmed the predominance of dust particles during a storm, with high AOD more than 1.0, using the MODerate-resolution 65 Imaging Spectroradiometer (MODIS) AOD dataset. Moreover, dust AOD (DAOD), namely the AOD for dust aerosols, can be retrieved using AOD and the simultaneous outputted parameters, such as Angstrom Exponent and sphericity. For instance, DAOD has been successfully retrieved using the MODIS DB aerosol product (Ginoux et al., 2010, 2012; Pu and Ginoux, 2018)."*

TO

*"Quantitative aerosol satellite remote sensing technology is expected to overcome the inherent limitations of ground-based networks by providing extensive and long-term datasets. These datasets are crucial for monitoring global and regional aerosol variations and tracking dust aerosol transport (Baddock et al., 2021; Chen et al., 2023a). Strictly verified Aerosol Optical Depth (AOD) datasets from various satellite sensors have been widely used for regional and global dust research. For example, Filonchyk (2018) utilized the MODerate-resolution Imaging Spectroradiometer (MODIS) AOD dataset and confirmed the predominance of dust particles during a storm, with high AOD exceeding 1.0. Furthermore, dust AOD (DAOD), which specifies the AOD attributable to dust, can be retrieved from AOD measurements and the simultaneously outputted parameters, such as the Angstrom Exponent and sphericity. For instance, DAOD has been successfully retrieved using the MODIS DB aerosol product (Ginoux et al., 2010, 2012; Pu and Ginoux, 2018)."* (Lines 61-69, page 3)

8. Line 129: "characterises"? "metrological"?

Response 8:

This sentence has been removed.

9. Line 141: "CALOP"?

Response 9:

Thanks for your careful checks. We are sorry for our carelessness. In our revised manuscript, the typo is revised.

we have corrected the "CALOP" into "CALIOP".

10. Line 154: ", according to. "?

  (Line 154)

Response 10:

This sentence has been replaced with

FROM

*"To facilitate a comprehensive analysis, the mainland China has been divided into four sub–regions: the Northwest, Qinghai–Tibet, the North, and South (Fig. 1), according to."*

TO

*"Based on the administrative division of China, we divide the research area into six sub-regions: Northwest China (NWC), North China (NC), Northeast China (NEC), Southwest China (SWC), Central South China (CSC), and East China (EC) as shown in Figure 1."* (Lines 142-145, page 5)

11. Line 197: "from 2001 to the present" – when is "present"?

12. Line 198: "0.05o×0.05o"?

Response 11-12:

We apologize for the typos in the manuscript. We have removed the description of the MODIS IGBP land cover product in satellite data as we are not discussing the variations in annual mean DAOD over six main land covers.

13. Line 278: "NECP"??

Response 13:

we have corrected the "NECP" into "NCEP".

---

## Author Comment (AC2)

Response to Reviewer 2 Comments

Dear reviewer:

Thanks very much for taking your time to review this manuscript. We really appreciate all your valuable comments and suggestions! We have carefully addressed each comment in blue with the original comments in black. Changes are highlighted in yellow in the revised manuscript.

General comments:

Multiple dust aerosol related datasets of MERRA-2, OMI and CALIPSO were used in this manuscript to study the spatiotemporal distribution of dust in China. Also, the dust transport and sources were analyzed using air mass trajectory statistic methods. Although the manuscript includes many valuable data and statistic results, it lacks one or more focused scientific points which leading the data analysis and most results are basic statistics from the data directly. The spatiotemporal characteristics of China dust have been investigated in many previous studies, what are the new scientific discoveries in this study? In my opinion, the manuscript needs to be rearranged focused one or two scientific points with deeper data analysis and scientific discussion.

Response:

First of all, we agree with you on that the spatiotemporal characteristics of dust aerosols in China have been investigated in many previous studies. These studies offer excellent support for the advancement of our work. Our study utilizes MERRA-2 and CALIPSO data to analyse the four-dimensional (latitude, longitude, altitude, and time) features of dust aerosols. Additionally, the distinctiveness of this study lies in its emphasis on the sources and transport pathways of dust for 16 selected cities (13 provincial capitals and 3 dust source cities) in the "Three-North" region. To make it clearer, we have revised the final paragraph of the introduction to underscore the paper's aim, which is to conduct

a more comprehensive analysis of the spatial and temporal dynamics and trajectories of dust in this region. The new scientific discoveries in this study are as follows:

(1) Regarding the four-dimensional analysis of dust storm changes, we have noticed an increasing trend in dust storms during autumn in the northwest region in recent years, as shown in Fig. R1 (Fig. 3b in the revised version). This is in contrast to the downward trend in other three seasons which is more consistent to previous studies. Exploration of the autumn trend anomaly is a challenging task due to the lack of ground-based DAOD observation data for MERRA-2 DAOD validation in the northwest region. We here instead analyzed the seasonal horizontal visibility in this area to confirm the observed trend. The results indicate that from 2007 to 2021, there is a general downward trend in the autumnal average visibility in the northwest region, as depicted in Fig. R2a. This decrease is very likely to be attributed to the increased presence of dust aerosols, which is reflected in the increased DAOD. Additionally, a significant correlation is observed between DAOD and visibility (Fig. R2b). The distribution of stations used to calculate the seasonal average visibility in the northwest is illustrated in Fig. R3. Nonetheless, the underlying causes of this phenomenon remain to be elucidated, and future research will necessitate a comprehensive analysis that incorporates additional datasets to discern the driving factors behind it. Our findings should stimulate further research in this area.

[Figure]

Fig. R1: Interannual variation of seasonal mean DAOD in northwest China.

[Figure]

Fig. R2: Interannual variation of Autumn mean visibility in northwest China (a) and the scatterplot of Autumn DAOD means vs. visibility means (b).

[Figure]

Fig. R3: Distribution of visibility sites.

(2) Regarding the trajectory analysis of dust storms, most studies typically examined a limited number of events or focus on a few selected sites. Instead, we have undertaken a more comprehensive analysis that spans a seven-year timeframe, encompassing the origins and trajectories of all dust storms across 16 representative cities in the entirety of Northwest China (NWC), North China (NC), and Northeast China (NEC), aka "Three-North" region. Our findings reveal substantial variability in the sources and trajectories of dust storms within the same region, with distinct patterns emerging across different seasons. For instance:

(a) The air mass trajectories with the highest proportion reaching Lanzhou originate from different dust sources depending on seasons: the Tengger Desert during spring (48.86%,), the Badain Jaran Desert during summer

and winter (52.67% and 49.69%, respectively), and Hexi Corridor during autumn (42.46%) . Western Inner Mongolia and the Hexi Corridor are the major dust sources with extremely high WPSCF values in winter and spring, while in winter dust source was more concentrated around the Tengger Desert. In the other two seasons, dust sources remain consistent in these two regions but with lower WPSCF values.

(b) During spring in Hohhot, high WPSCF values are widely distributed around the Gobi Desert in Inner Mongolia and Mongolia. From autumn to winter, the dust sources become more concentrated around the Chinese part of the Gobi Desert.

(c) In the provincial capital cities of Northeast, potential dust sources during spring are widespread but are particularly concentrated around the Horqin Sandy Land in all seasons except summer, indicated by high WPSCF values.

(3) Furthermore, several dust source analysis studies used long-term CALIPSO satellite aerosol products, which has much lower observation frequencies compared to those from ground stations and may lead to the omission of some dust events. In contrast, our study leverages PM measurements from ground stations, which provide an hourly temporal resolution. The seven-year time span minimizes the impact of statistical randomness on our findings, thereby facilitating a comprehensive and detailed source-trajectory analysis of dust storms.

Specific comments:

1. Figure 1, Inner Mongolia should not be included in Northwest area.

Response :

Thank you for your reminder. In the revised manuscript, we have reorganized the study area into six sub-regions based on China's administrative divisions (Song et al., 2021; Liu, et al., 2022; Wang et al., 2023): Northwest China (NWC), North China (NC), Northeast China (NEC), Southwest China (SWC), Central South China (CSC),

and East China (EC). In accordance with this new classification, Inner Mongolia is included in North China, as shown in Fig. R4 (Fig.1 in the revised version). Additionally, we have made corresponding revisions to Section 2.1, which introduces the study area, along with the analytical sections, 4.1 and 4.2, to ensure they are in harmony with these refined regional divisions.

[Figure]

Fig. R4: The study area (a) and the main deserts in the study area (colored in b), with the distribution of Ambient air quality monitoring stations (dots in a)

Li, Y. R., Niu, Y., Wei, T. X., Liang, Y. S., Chen, P., Ji, X. D., and Zhang, C. J.: Mapping rainfall interception for assessing ecological restoration sustainability in China. Environ. Res. Lett., 7(10): 104007, https://doi.org/10.1088/1748-9326/ac8605, 2022.

Song, Z. H., Xia, J., She, D. X., Li, L. C., Hu, C., Hong, S.: Assessment of meteorological drought change in the 21st century based on CMIP6 multi-model ensemble projections over mainland China, J. Hydrol., 601: 126643, https://doi.org/10.1016/j.jhydrol.2021.126643, 2021.

Wang, Y., Xu, T. T., Shi, G. M., Yang, F. M., Tang, X. L., Zhao, X. L., Wan, C. Y., Liu, S. L.:Climatology of the planetary boundary layer height over China and its characteristics during

periods of extremely temperature, Atmos. Res., 294, 106960, https://doi.org/10.1016/j.atmosres.2023.106960, 2023.

2. For PSCF dust source analysis, how to distinguish the contribution of local emitted air pollution and long-range transported dust aerosol?

Response:

The PSCF does not directly differentiate between local sources and long-range transported dust aerosols; it merely calculates the probability of each grid cell being a dust source based on backward trajectories produced by the HYSPLIT model. To calculate the PSCF, the study area was divided into $0.5° \times 0.5°$ grid cells. $PSCF_{ij}$ is defined as: $PSCF_{ij} = m_{ij}/n_{ij}$, where $n_{ij}$ is the total number of backward trajectory endpoints that fall in the ij-th cell, and $m_{ij}$ is the number of backward trajectory endpoints in the same cell with PM10 exceeding the criterion value. Considering that our initial explanation of the PSCF may have been misleading, we have rewritten section 3.2.3 in the introduction to provide a more accurate and detailed explanation of the PSCF methodology.